# DNA damage drives antigen diversification in *Trypanosoma brucei*

Jaclyn E. Smith[1], Kevin J. Wang[1], Erin M. Kennedy[1,4], Jane C. Munday[2], Lulu Singer[1], Jill M. C. Hakim[1], Jaime So[1], Alexander K. Beaver[1,3], Aishwarya Magesh[1], Shane D. Gilligan-Steinberg[1,5], Jessica Zheng[1], Bailin Zhang[1,6], Dharani Narayan Moorthy[1], Zachary E. Brown[1], Elgin Henry Akin[1], Lusajo Mwakibete[1], Richard McCulloch[2] & Monica R. Mugnier[1✉]

Antigenic variation, using large genomic repertoires of antigen-encoding genes, allows pathogens to evade host antibody. Many pathogens, including the African trypanosome *Trypanosoma brucei*, extend their antigenic repertoire through genomic diversification. Although evidence suggests that *T. brucei* depends on the generation of new variant surface glycoprotein (VSG) genes to maintain a chronic infection[1–4], a lack of experimentally tractable tools for studying this process has obscured its underlying mechanisms. Here we present a highly sensitive targeted sequencing approach for measuring VSG diversification. Using this method, we demonstrate that a Cas9-induced DNA double-strand break within the VSG coding sequence can induce RAD51- and BRCA2-dependent VSG recombination with patterns identical to those observed during infection. These newly generated VSGs are antigenically distinct from parental clones and thus capable of facilitating immune evasion. Together, these results provide insight into the mechanisms of VSG diversification and an experimental framework for studying the evolution of antigen repertoires in pathogenic microorganisms.

Pathogen survival in a host depends upon effective and continuous immune evasion. Several bacteria and eukaryotic pathogens have adopted the strategy of antigenic variation to evade host immunity, a process in which the pathogen continuously alters antigenic surface proteins to escape the host's adaptive immune response. The African trypanosome *Trypanosoma brucei*, a unicellular eukaryotic parasite and causative agent of human and animal African trypanosomiasis, uses an especially sophisticated system of antigenic variation. The parasite, which remains extracellular throughout infection and thus faces a perpetual onslaught of host antibody, periodically 'switches' expression of a surface coat consisting of $10^7$ copies of a single, immunogenic protein known as the variant surface glycoprotein (VSG). This process allows parasites to escape host antibody and maintain a chronic infection.

Although the *T. brucei* VSG repertoire contains thousands of VSGs, it is probably too small to maintain a chronic infection through VSG switching alone. During an infection, each *T. brucei* parasite expresses a single VSG at a time from one of about 15 telomeric bloodstream expression sites (BESs) (the 'active' BES)[5]. The remaining VSG-encoding genes are stored in other expression sites, subtelomeric arrays and minichromosomes, all of which remain transcriptionally silenced[6]. The parasite switches its VSG either by transcriptional activation of a silent BES (in situ switching) or through a gene conversion event in which a new VSG is copied into the active expression site. Although gene conversion-based switching allows for the activation of VSGs outside of a BES, analysis of the *T. brucei* genome has shown that only around 20% of the VSGs in the parasite genome are full-length genes encoding a functional VSG protein. The remaining approximately 80% of VSGs in the parasite genome consist of pseudogenes or gene fragments[6–8] and cannot immediately be used for immune evasion through in situ or gene conversion switching. Moreover, the number of VSGs expressed in a population of parasites at a single time during experimental infection sometimes exceeds the total number of intact VSGs in the parasite genome[1,9], further indicating that the repertoire of intact VSGs is insufficient to achieve the antigenic diversity required to maintain a chronic infection.

Evidence suggests that *T. brucei* deals with this shortage of antigens through diversification of the VSG repertoire. Many studies of experimental infections in mice have shown that novel VSGs, generated during infection, predominate at later stages of infection[1,2,4], whereas analysis of parasites from natural human infections revealed expressed VSGs that were nearly completely absent from the genomes of contemporary field isolates[10]. These observations suggest that the generation of new VSGs has a critical role in sustaining *T. brucei* antigenic variation[3].

There are two mechanisms thought to be responsible for extending the VSG repertoire: mosaic formation and de novo point mutation. Mosaic VSGs form when two or more VSG genes combine through segmental gene conversion to form a novel VSG. This mechanism allows parasites to access pseudogenes and VSG fragments within the

[1]W. Harry Feinstone Department of Molecular Microbiology and Immunology, Johns Hopkins Bloomberg School of Public Health, Baltimore, MD, USA. [2]University of Glasgow Centre for Parasitology, School of Infection and Immunity, University of Glasgow, Glasgow, UK. [3]Department of Pathology, Johns Hopkins School of Medicine, Baltimore, MD, USA. [4]Present address: Department of Molecular Microbiology and Immunology, Brown University, Providence, RI, USA. [5]Present address: Department of Bioengineering, University of Washington, Seattle, WA, USA. [6]Present address: Scripps Research Department of Integrative Structural and Computational Biology, La Jolla, San Diego, CA, USA. ✉e-mail: mmugnie1@jhu.edu

repertoire. Where mosaic VSGs have been described in the literature, they are often found under strong, antibody-mediated selection[11–15] or late during infection[1,4,16,17], making it difficult to discern how exactly they arose. VSGs also appear occasionally to acquire de novo point mutations[12,18], although these can be difficult to distinguish from small gene conversion events. Ultimately, de novo mutation of VSGs would allow parasites to generate new VSG sequences regardless of the contents of their repertoire, further amplifying diversity.

Despite its clear importance, the mechanisms driving VSG diversification, whether by mutation or recombination, remain poorly understood. It is plausible that DNA damage and repair may have a role in either mechanism, as expressed VSG genes sit between two highly repetitive[6,19] and damage-prone[20,21] stretches of DNA, the conserved 70-base pair (bp) repeat and the telomere. Locus-directed mapping has detected DNA breaks within the 70-bp repeats at the active expression site[20] and near telomeres within both the active BES[21] and silent BESs[21], whereas genome-wide DNA break mapping has shown highly abundant, complex breaks within the expressed VSG and confined to the active BES[22]. Experimental evidence implicates DNA damage in VSG switching more generally, as a DNA double-strand break induced upstream of the VSG[20,21] and within the first 15 bases of the VSG-2 coding sequence[23] induces a switch. One recent study showed that a DNA double-strand break in the coding sequence of a VSG can generate new mosaic VSGs[24], but it is unclear whether this mechanism reflects the patterns of mosaic formation in vivo; the molecular mechanisms driving this recombination are unknown.

Here we present a comprehensive toolkit for the controlled and reproducible study of the diversification of individual VSGs. Using a barcode-based targeted RNA sequencing approach, we show that DNA double-strand breaks can trigger the formation of mosaic VSGs that are identical to those observed in vivo during infection. In addition to identifying a potential hypervariable region within the VSG protein, our experimental approach revealed both the sequence requirements and key cellular machinery needed for antigen diversification in *T. brucei*.

## DNA breaks trigger mosaic VSG formation

A number of studies have suggested that VSG diversification occurs, at least some of the time, within the active VSG expression site[1,18]. For this reason, we focused our analyses on diversification of the actively expressed VSG. Because a double-strand DNA break upstream or within the first 20 bp of the VSG in the active BES is known to induce switching[20,21,23], we hypothesized that a break within the VSG coding sequence might result in mosaic VSG formation. To investigate this, we engineered tetracycline-inducible Cas9-expressing EATRO1125 parasites, which express the VSG AnTat1.1 from the natural active expression site, and induced breaks across the AnTat1.1 coding sequence using a set of guide RNAs.

To evaluate potential recombination outcomes, we developed VSG anchored multiplex PCR sequencing (VSG-AMP-seq), a technique that overcomes previous obstacles to studying VSG diversification by providing both high-throughput and highly accurate sequences (Fig. 1). Our approach uses long unique molecular indexes to generate high-confidence consensus sequences[25]. Consolidating reads into consensus sequences allows errors such as PCR chimeras, which occur during later cycles of PCR and therefore represent a minority of sequences in a consensus group[26], to be eliminated while true events are retained. Owing to its selective, target-specific amplification, this method can sensitively detect thousands of rare diversification events, which we validated through VSG-AMP-seq analysis of parasite populations mixed together in known proportions (Extended Data Fig. 1a,b).

To induce breaks across AnTat1.1, we induced Cas9 expression for 24 h and then transfected in DNA amplicons containing a T7 promoter and a guide RNA targeting various regions throughout the AnTat1.1

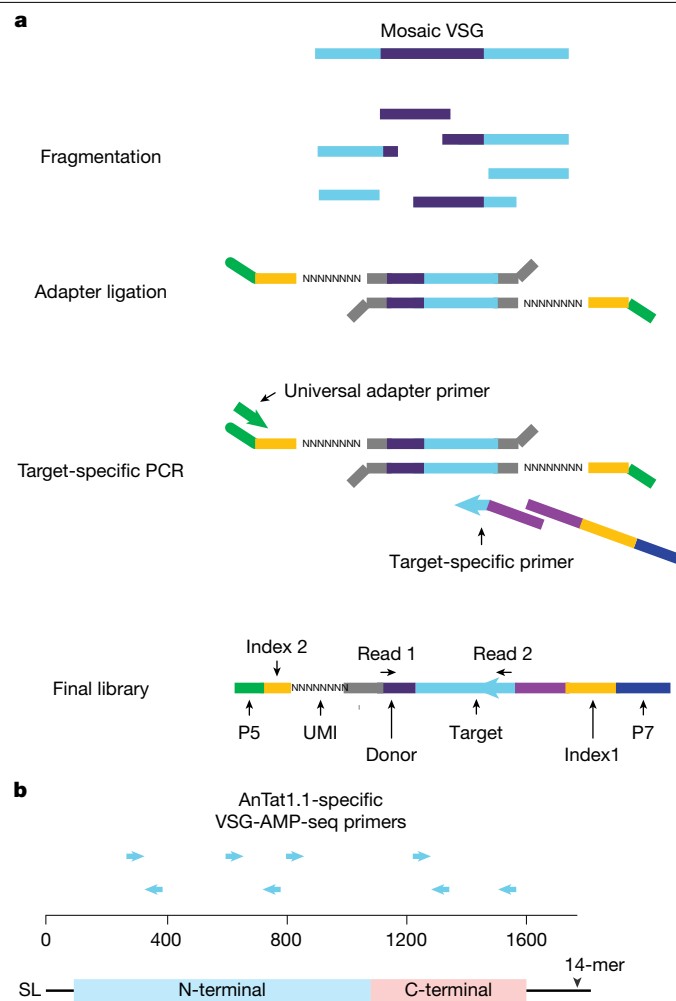

**Fig. 1 | VSG-AMP-seq protocol. a**, Schematic of the library prep for VSG-AMP-seq. **b**, Locations of target-specific primers for VSG AnTat1.1. Panel **a** adapted from ref. 59, Springer Nature America. SL, 5′ splice leader sequence; 14-mer, 3′ sequence conserved in all VSG transcripts; P5, P5 universal illumina adapter; P7, P7 universal illumina adapter; UMI, unique molecular index.

coding sequence to induce breaks in the VSG. Parasites were collected 2 days after transfection of the guide, and mosaic derivatives of AnTat1.1 were analysed by VSG-AMP-seq (Fig. 2a and Extended Data Figs. 2a,b, 3a,b and 4b). We detected thousands of recombination events from two independently generated Cas9 clones (C1 = 5,956, C2 = 4,488).

Our analysis showed diverse mosaic recombination events centred around each break site. Such events were virtually absent from the negative (no guide) control, suggesting this process does not occur at a high rate in the absence of a trigger or in the presence of Cas9 alone. Notably, as the DNA breaks progressed further away from the centre of the AnTat1.1 coding sequence, the frequency of observed mosaics decreased markedly (Fig. 2b). This did not appear to be related to the guide sequences (Extended Data Fig. 2c–g) or to guide cutting efficiency (Extended Data Fig. 2h,i). Rates of parasite death and VSG switching after induction of DNA breaks also did not vary significantly between guides (Extended Data Fig. 2j,k). To ensure that the observed mosaic events represented mosaic VSGs that were truly expressed by the parasite, and not technical artefacts or VSGs incapable of being stably expressed by *T. brucei*, we also obtained individual clones of parasites expressing AnTat1.1-derived mosaic VSGs (Extended Data Figs. 2a and 4c). These parasites expressed VSGs containing recombination events identical to those observed using VSG-AMP-seq. These results indicate that DNA damage within the active VSG can trigger

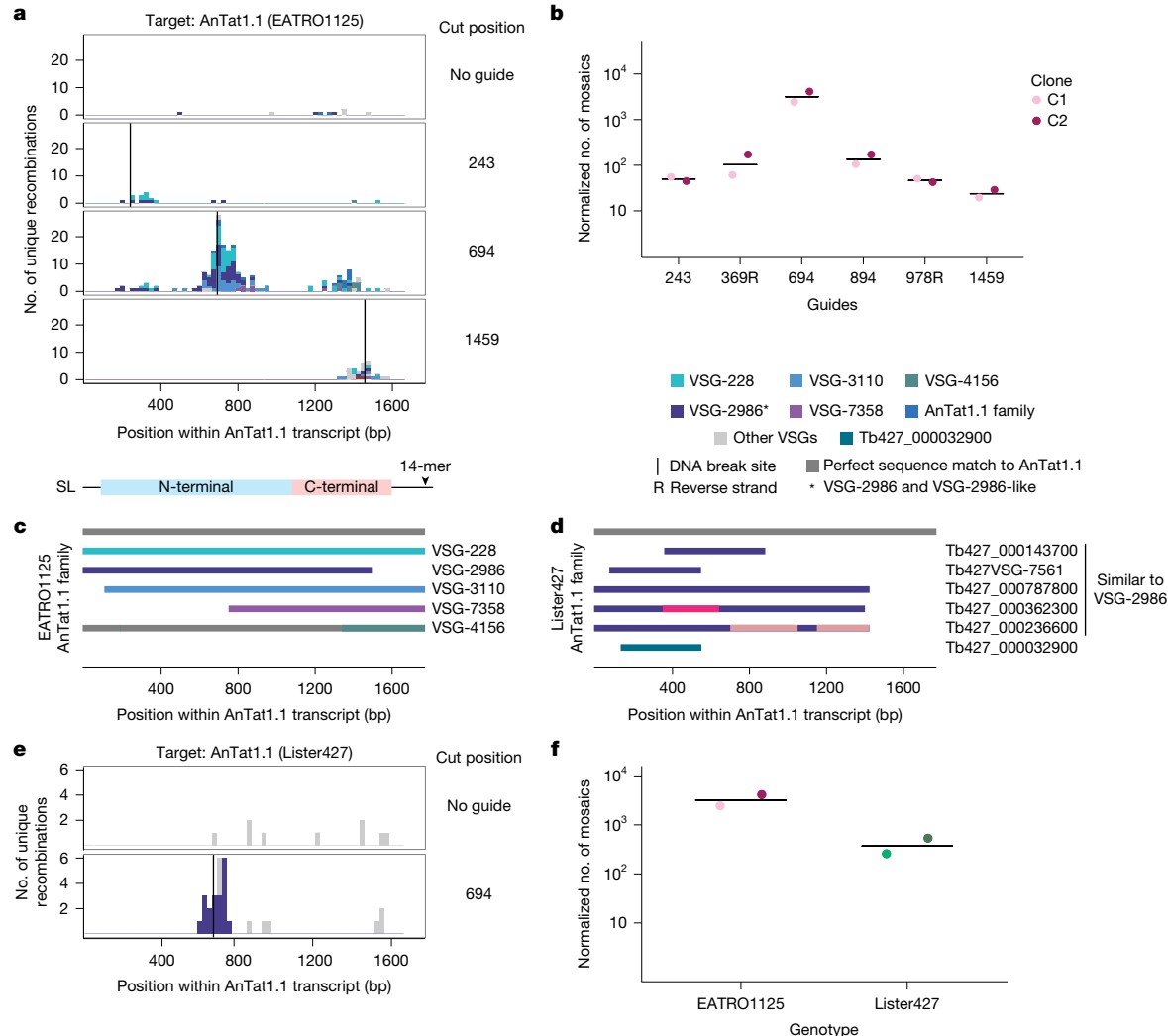

**Fig. 2 | DNA double-strand breaks trigger mosaic VSG formation when homology is available. a**, Histogram of unique recombination events identified along the AnTat1.1 transcript. The Cas9 DNA break site is indicated by a vertical line. Cut positions shown, relative to the VSG transcript 5′ end, are: 243, 694 and 1459. Histogram colours indicate the donor VSG identified. The midpoint of the perfect homology between AnTat1.1 and the donor VSG at the recombination site is plotted. If a mosaic sequence matched more than one potential donor VSG, the average recombination position was plotted. **b**, Quantification of break-induced mosaic recombination events. Cut positions, relative to the VSG transcript 5′ end, are: 243, 369, 694, 894, 978 and 1459. R indicates a guide that binds to the reverse strand. The number of recombination events detected within 250 bp up- or downstream of the cut site was normalized to the number of total unanchored reads aligning within that region compared with the unanchored read count from the region with the lowest coverage to control for sequencing depth (*n* = 2, two independent clones). Mean represented by a horizontal line. **c**, Schematic of the AnTat1.1 family aligned to the AnTat1.1 transcript. **d**, Schematic of the AnTat1.1 family members in the Lister427 strain. **e**, A histogram of unique recombination events identified within AnTat1.1-expressing Lister427 parasites after a cut at position 694 along the AnTat1.1 transcript, plotted as in **a**. **f**, Quantification of mosaic recombination events induced by Cas9 at position 694. EATRO data are from **b** (*n* = 2, two independent clones). Mean represented by a horizontal line.

the formation of mosaic VSGs, with recombination events centred around the site of DNA damage.

## Homology drives mosaic VSG formation

Analysis of the mosaic recombination events detected by VSG-AMP-seq revealed an important role for sequence homology in the formation of mosaic VSGs. Within the isolated mosaic clones, we observed short insertions (less than 300 bp, average = 55 bp, median = 29 bp) predominating among the events (Extended Data Fig. 4d). Although read lengths limited our ability to detect larger insertions with VSG-AMP-seq, approximately 55–60% of the mosaic reads detected by VSG-AMP-seq contained the same short insertions (average C1 = 46 bp, average C2 = 45.8 bp) (Extended Data Fig. 4e). Almost all (C1 = 99.71%, C2 = 99.46%) of these recombination events occurred at a region

of shared sequence between AnTat1.1 and each donor VSG, with an average length of approximately 9 bp (average length: C1 = 9.13 bp, C2 = 9.44 bp, median = 6 bp) (Extended Data Fig. 4a,f). Notably, only a small number of donor VSGs were used for most recombination events. These donors are members of a 6-VSG family that contains AnTat1.1 and represent the only sequences within the EATRO1125 VSGnome with substantial homology to this VSG (Fig. 2c). N-terminal recombination events appear restricted to just these family members (C1 = 100%, C2 = 99.87%). We observed the same short insertions when breaks were induced in the coding sequence of two other VSGs with homologous family members, Tb427VSG-8 and EATRO1125VSG-73, suggesting the observed patterns are not specific to AnTat1.1 mosaics (Extended Data Fig. 5a–d; although Tb427VSG-8 is in the EATRO genome, it is annotated only in the Lister427 VSGnome[6] and therefore lacks an EATRO VSG identifier). Together, these results indicate that VSG sequence

homology influences the outcome of VSG recombination after a DNA break.

Not all VSGs share sequence homology with other members of the VSG repertoire, however. To determine the outcome following DNA damage of a VSG that has no potential homologous donors, we used the same Cas9 system in the commonly used Lister427 *T. brucei* line. We cut the actively expressed VSG, VSG-2, which lacks homologous family members, by inducing Cas9 expression for 7 days in parasites containing constitutively expressed guides targeting VSG-2. Parasite clones isolated after a break in VSG-2 no longer expressed VSG-2, and the VSGs expressed by these clones, which were all expression site-associated, showed no evidence of mosaic recombination (Extended Data Fig. 4g; cut position 707, *n* = 4; cut position 1082, *n* = 3). This further supports the hypothesis that sequence homology between the parent and donor VSGs is required for mosaic VSG formation.

We sought to determine what proportion of the VSG repertoire contains VSGs that are members of VSG families and thus capable of diversifying through break-induced mosaic formation. We found that most (more than 75%) of the known VSG sequences are capable of diversifying through the mechanisms described here (Extended Data Fig. 4h).

## Mosaic formation is templated

Although most AnTat1.1-derived mosaics were identical to the putative donor VSG, many of these insertions altered only a few bp in AnTat1.1. We thus reasoned that it was possible that these events were de novo mutations created during DNA repair that happened to match the putative genome-encoded donor VSG. To test this possibility, we took advantage of the unique VSG repertoires in the EATRO1125 and Lister427 parasite strains[6,27]. Although AnTat1.1 is not endogenously present in Lister427 parasites, there are five VSGs nearly identical to the AnTat1.1 family member VSG-2986 that could serve as donor VSGs for the formation of AnTat1.1-derived mosaic VSGs in Lister427 parasites (Fig. 2d). We thus engineered dox-inducible Cas9-expressing Lister427 parasites and replaced VSG-2 with AnTat1.1 at the active expression site. We then induced breaks in AnTat1.1 as before and detected hundreds of recombination events in two independent clones (L1-A1 = 398, L1-A2 = 833; Fig. 2f). All mosaic recombination events detected used donor VSGs found exclusively in the Lister427 genome (Fig. 2e), indicating that the sequence changes observed after break induction are templated and not the result of de novo mutation.

## Donors can be used throughout the genome

We wondered whether the location of a VSG within the genome could influence its use as a donor for mosaic recombination. To evaluate the role genomic context has in the selection of donor VSGs, we again took advantage of the unique repertoires of the EATRO1125 and Lister427 strains[6,27]. Using the same AnTat1.1-expressing Lister427 Cas9 parasites, we inserted an EATRO1125 VSG, VSG-228, which is commonly used as a donor in AnTat1.1 mosaics and is absent from Lister427 parasites, into three genomic locations: the minichromosomes, where VSGs are typically found, and the ribosomal DNA (rDNA) spacer and the tubulin array, where VSGs are not typically stored (Fig. 3a and Supplementary Fig. 1). We then induced breaks at position 694 in AnTat1.1 and analysed donor selection by VSG-AMP-seq.

The endogenous Lister427 donors were selected at the same frequency in all samples, suggesting that the presence of an extra donor did not interrupt the formation of these mosaic VSGs (Fig. 3b). The inserted donor, VSG-228, was used regardless of its location (Fig. 3b and Extended Data Fig. 6a). Notably, only the coding sequence of VSG-228 was inserted into the Lister427 background. This indicates that neither the upstream 70-bp repeats typically found interspersed within the VSG archive[7,8,28,29] nor the 5'- and 3'-untranslated regions

(UTRs) of the VSG are required for locating donor templates. Taken together, these data suggest that parasites possess a homology search mechanism that allows donor VSGs to be used regardless of genomic location.

## Imperfect flanking homology is required

To further understand the sequence requirements for mosaic VSG formation, we inserted truncations of VSG-228, ranging in length from 500 bp to 100 bp, into the rDNA spacer of AnTat1.1-expressing Lister427 parasites. Each truncated donor was centred around the break site at position 694, except for the '100-bp offset' donor, which was designed to reflect the boundaries of a 100-bp sequence that is commonly inserted into AnTat1.1 (Extended Data Fig. 6e). Evaluation of mosaic formation by VSG-AMP-seq showed that donors as short as 200 bp were used as efficiently as the full-length VSG (Fig. 3c and Extended Data Fig. 6b,c). A 100-bp donor sequence centred around the break site could be used with lower efficiency than the full-length VSG, whereas no recombination was observed in the '100-bp offset' parasites (Extended Data Fig. 6d). These data suggest that mosaic recombination requires less than 50 bp of flanking homology around the break site, and only 100 bp of flanking homology is required for recombination with similar efficiency to a full-length donor.

Finally, we used this system to further confirm that mosaic formation is a templated gene conversion event and not the result of crossing-over. Using J1339 Cas9 TetR T7RNAP Lister427 parasites expressing AnTat1.1 containing an inserted VSG-228 donor and 694 guide in the genome, we isolated mosaic-expressing parasite clones after break induction, selected clones that used VSG-228 as the donor and evaluated the silent VSG-228 donor by nanopore amplicon sequencing (Extended Data Fig. 7). In all cases, the donor VSG was intact in the genome after mosaic formation, demonstrating that during mosaic formation the donor VSG serves only as a template for DNA repair.

## Mosaics form by homologous recombination

Given the homology requirements of mosaic recombination, we wondered whether factors known to be involved in homologous recombination are required for break-induced mosaic formation. Both RAD51 (ref. 30) and BRCA2 (ref. 31) are known to have a role in DNA repair and antigenic variation in *T. brucei*. In addition, microhomology-mediated end-joining, which relies on around 5–20 bp of imperfect homology[32], comparable to that which we observe flanking mosaic recombination sites, has been described in *T. brucei* and occurs independently of RAD51 (refs. 32–34). To test the role of RAD51 and BRCA2 in mosaic recombination, we transiently transfected amplicons encoding single guide RNAs (sgRNAs) targeting position 694 in Cas9-expressing parasites lacking either RAD51 or BRCA2 (Extended Data Fig. 8a). Mosaic recombination was markedly reduced in the absence of BRCA2 and completely abolished in the absence of RAD51 (Fig. 3d,e). Together, these results indicate that break-induced mosaic recombination requires RAD51 and BRCA2 to operate.

## In vitro mosaics mimic natural mosaics

To investigate whether the mosaic VSGs we detected after break induction in vitro reflected the events that occur naturally in vivo, we performed VSG-AMP-seq on parasites isolated from wild-type (WT) mouse infections 15 days after infection, when AnTat1.1 has been mostly eliminated from the blood (Extended Data Fig. 9b,c). We observed a strong C-terminal bias to all recombination events (316 recombination events across 7 mice). Very few sequences aligned to AnTat1.1 within the N terminus, suggesting that most of these mosaic VSGs were complete replacements of the VSG N-terminal domain (Fig. 4a and Extended

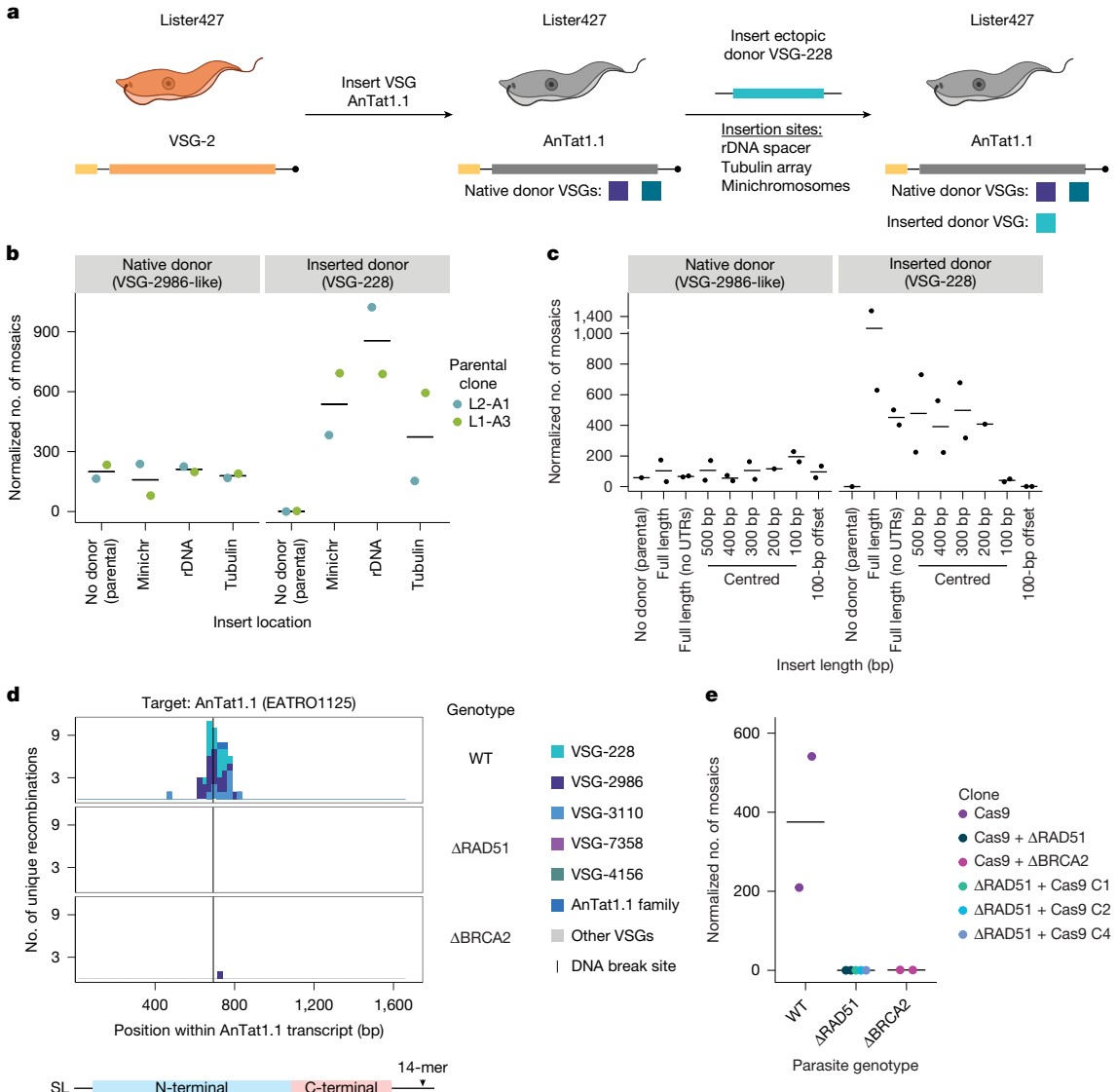

**Fig. 3 | Mosaic recombination occurs via homologous recombination.**
**a**, Schematic of the generation of Lister427 parasites expressing AnTat1.1 and harbouring a silent copy of VSG-228. **b**, Quantification of mosaic recombination using VSG-2986-like or VSG-228 as a donor following induction of Cas9. The number of recombination events detected within 250 bp up- or downstream of position 694 was normalized to the number of total unanchored reads aligning within that region compared with the unanchored read count from the region with the lowest coverage. *n* = 2, each parental clone was independently generated. Mean represented by a horizontal line. **c**, Quantification of mosaic recombination using VSG-2986-like or VSG-228 as a donor following induction of Cas9 in AnTat1.1-expressing Lister427 parasites carrying truncations of VSG-228 inserted into the rDNA spacer. The number of recombination events

detected was quantified and normalized as in **b**. *n* = 2, unique clones were isolated for each inserted VSG-228 donor from the same parental. Mean represented by a horizontal line. **d**, Histogram of unique recombination events after a cut at position 694 in AnTat1.1 in EATRO1125 WT parasites or parasites with knockouts of RAD51 or BRCA2. **e**, Quantification of mosaic recombination events induced by Cas9 at position 694. *n* = 2 biological replicates using the same clone; *n* = 5 for ΔRAD51, 2 replicates from the same clone and 3 uniquely derived clones. Clone names indicate the order in which each genetic modification occurred. The number of recombination events detected was quantified and normalized as in **b**. Mean represented by a horizontal line. Illustrations in **a** created in BioRender; Smith, J. https://biorender.com/4ub3zqc (2026).

Data Fig. 9a). These donor VSGs shared significantly less homology with AnTat1.1 than the AnTat1.1 family members, usually sharing spans of only 100 bp or less of imperfect homology within the C-terminal region of the VSGs. Nevertheless, there was still a short (average = 12 bp) span of perfect identity between AnTat1.1 and the donor at the recombination site (Fig. 4e). Because VSGs expressed by parasites at the second peak of parasitaemia are antigenically distinct from AnTat1.1, we reasoned these N-terminal replacement VSGs ensured complete immune evasion as only a small portion of AnTat1.1 was retained.

We wondered whether the restriction of events to the VSG C terminus reflected a mechanistic bias towards recombination within the VSG C-terminal domain or the effect of host antibody selection, with

only the most immune evasive recombination events surviving. To investigate these possibilities, we repeated the experiment in μMT mice[35], which do not have mature B cells and therefore do not generate antibodies. We again analysed parasites 15 days after infection, although AnTat1.1 is never cleared from these infections and parasitaemia remains high throughout (Extended Data Fig. 9b,c). In μMT mice, mosaic recombination events span the full length of the VSG (Fig. 4a and Extended Data Fig. 9a) (2,144 recombination events across 7 mice). In vivo, we observe short insertions (average = 41 bp) using donors homologous to AnTat1.1 flanked by short (average = 13 bp, median = 9 bp) regions of perfect identity, matching the patterns we observed in vitro (Fig. 4e,f). We also observed similar short mosaic

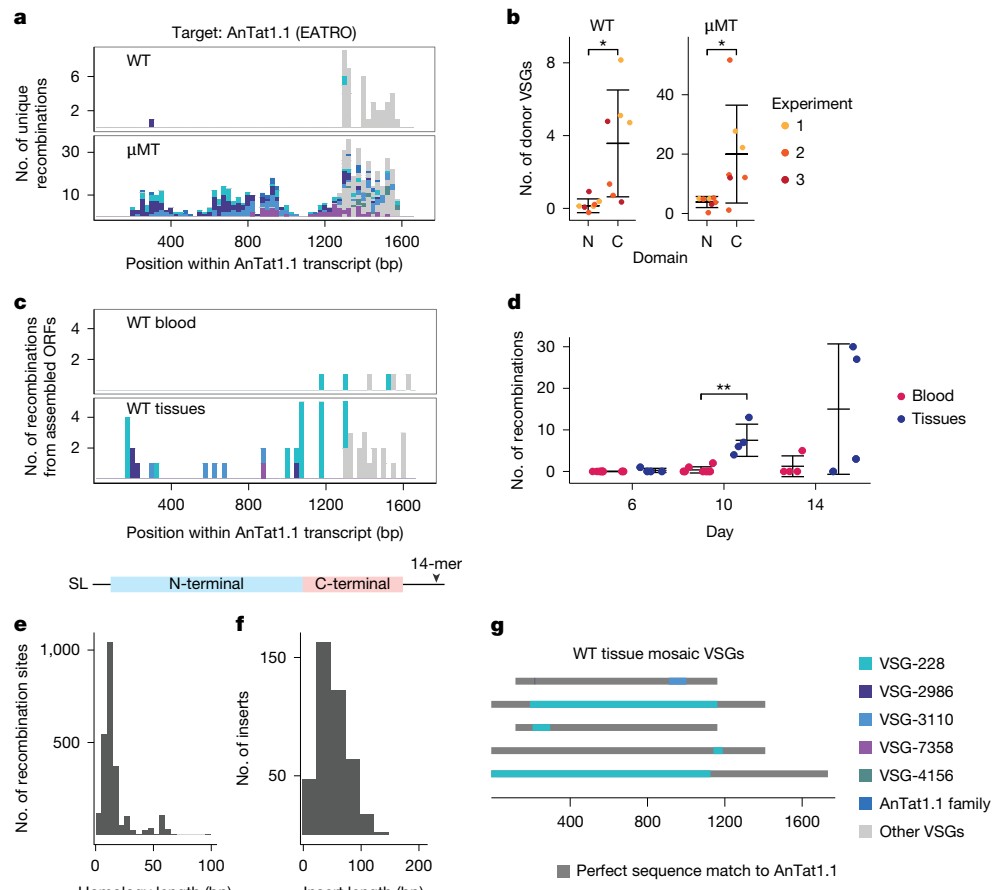

**Fig. 4 | AnTat1.1 mosaic VSGs form in vivo and are most prevalent in extravascular spaces. a**, Histogram of unique recombination events identified in all WT or μMT mice 15 days after infection. Recombination events found in multiple mice are represented once (WT $n = 7$; μMT $n = 7$, from 3 independent experiments). The midpoint of the perfect homology between AnTat1.1 and the donor VSG at the recombination site is plotted. If a mosaic sequence matched more than one potential donor VSG, the average recombination position was plotted. **b**, Quantification of the number of donor VSGs used in mosaic recombination events within the N- and C-terminal domains of AnTat1.1 in WT and μMT mice from **a**. Statistical significance was determined by a Shapiro–Wilk normality test followed by a two-tailed pairwise Wilcoxon signed-rank test (*$P < 0.05$, WT: 0.03351, μMT: 0.03429). Mean ± s.d. **c**, A histogram of the mosaic recombination events identified from VSG ORFs assembled in ref. 9. If a mosaic

recombination event was identified in more than one tissue within the same mouse, it was counted once ($n = 12$, 4 mice per tissue time point). **d**, Quantification of the mosaic recombination events detected in WT mouse blood or tissue. Statistical significance was determined by Shapiro–Wilk normality test followed by pairwise Wilcoxon rank-sum test within each time point (**$P < 0.01$, day 6: 0.1124, day 10: 0.004953, day 14: 0.2186). Mean ± s.d. **e**, Histogram of the length of the shared identity between AnTat1.1 and the donor VSG at the recombination sites. **f**, Histogram of donor VSG insertion lengths identified within individual reads. The insert length includes only newly inserted sequence and does not include recombination sites. **g**, Representative schematics of mosaic VSGs from assembled ORFs in ref. 9. ORF, open reading frame.

insertions in Tb427VSG-8 during μMT infections (average = 70.5 bp, median = 42 bp) using nanopore sequencing (Extended Data Fig. 5e–g).

Notably, as in the WT infections, a subset of recombination events in μMT mice occurring within the C terminus were not observed in vitro (Fig. 4b). We wondered whether the RAD51 null parasites, which were devoid of mosaic recombination in our in vitro system, could produce mosaic recombination events using an RAD51-independent mechanism in vivo. Infections in μMT mice initiated with ΔRAD51 parasites expressing AnTat1.1 contain a very small number of mosaic insertion events (Extended Data Fig. 8b–d). These data demonstrate that our in vitro system recapitulates a subset of in vivo recombination events, but there may be other pathways facilitating recombination in vivo in addition to those that can be triggered by a single, blunt DNA double-strand break.

## Mosaic VSGs form in extravascular spaces

Although parasites expressing AnTat1.1 are eliminated from the blood in WT mice by day 10, they persist within tissues until at least day 14 (ref. 9). Given that parasite clearance is delayed in extravascular spaces, we

wondered whether mosaic VSGs may be more prevalent in this parasite niche. To investigate this, we analysed all assembled VSGs from a previous study of extravascular parasite populations[9]. We found AnTat1.1 mosaics present in tissues identical to those observed in both μMT mice and in vitro (Fig. 4c,g). Moreover, we observe that mosaic derivates of AnTat1.1 increase over time within tissue spaces during infection (Fig. 4d). These data suggest that mosaic VSGs may accumulate within extravascular spaces, possibly owing to the slower VSG-specific parasite clearance in these spaces.

## Small VSG changes drive immune evasion

Many AnTat1.1-derived mosaic VSGs differ from their original sequence by only a few amino acids. To determine how these changes influence the antigenic character of the VSGs, we analysed live parasites expressing AnTat1.1 mosaic derivatives by flow cytometry using a potent rabbit anti-AnTat1.1 polyclonal antibody raised against purified AnTat1.1 protein[36] (Fig. 5a,b and Extended Data Fig. 10a) and mouse antisera collected after infection with AnTat1.1-expressing parasites (Fig. 5c and Extended

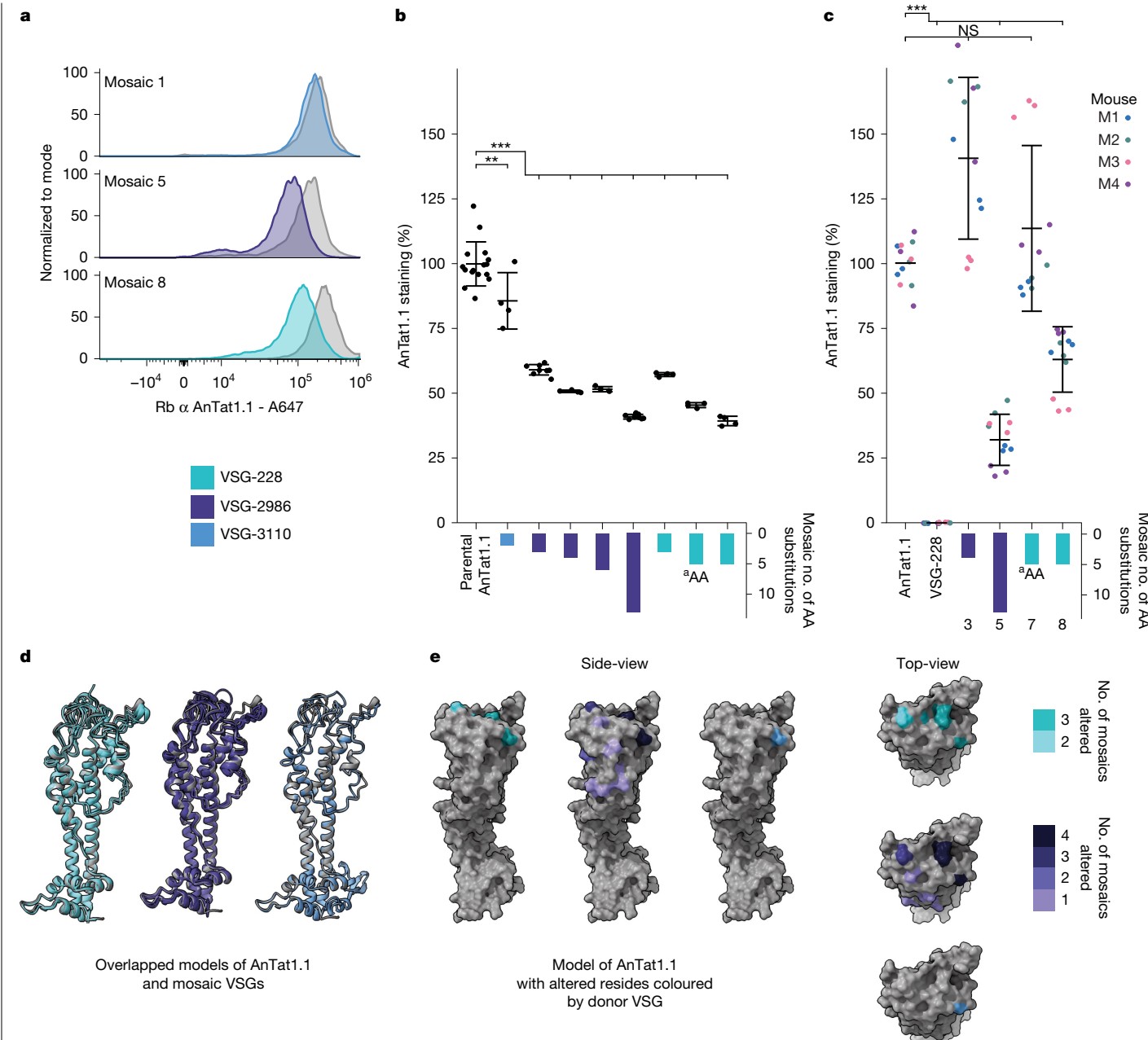

**Fig. 5 | Mutations in sequences encoding the top of the VSG alter antibody binding. a**, Representative histograms showing the binding of rabbit anti-AnTat1.1 antibody, as measured by anti-rabbit IgG Alexa Fluor 647 staining intensity, for parental controls (grey) and mosaic clones coloured by donor VSG. *n* = 4 technical replicates. **b**, Quantification of staining intensity changes for individual clones from **a** and Extended Data Fig. 10, based on median staining intensity. The median Alexa Fluor 647 intensities were normalized to the average of the parental clone. The numbers of amino acid substitutions are shown below the *x* axis for mosaics 1–8. *n* = 4 technical replicates. Statistical significance was determined based on a one-way ANOVA with a post hoc Tukey HSD (**\*\****P* < 0.01, \*\*\**P* < 0.001). Mean ± s.d. **c**, Quantification of staining intensity changes for a subset of individual clones stained with antisera from mice after infection with parasites expressing AnTat1.1, as shown by anti-mouse IgG Alexa Fluor 647

median staining. The median Alexa Fluor 647 intensities were normalized to the average AnTat1.1 staining with each mouse antibody. The numbers of amino acid substitutions are shown below the *x* axis in addition to the number associated with each mosaic. *n* = 3 technical replicates. *n* = 4 biological replicates of the mouse antisera. Statistical significance was determined based on a one-way ANOVA with a post hoc Tukey HSD (\*\*\**P* < 0.001). Mean ± s.d. **d**, Overlapping ribbon structures of AnTat1.1 and AnTat1.1 mosaic VSGs coloured by donor VSG as predicted by ColabFold. **e**, A space-filling model of AnTat1.1 highlighting the changed residues within the monomer for each donor VSG. Residues mutated in multiple mosaics are a darker shade. ᵃAA is a novel amino acid change not in AnTat1.1 or the donor, VSG-228. AA, amino acid; ANOVA, analysis of variance; HSD, honestly significant difference; NS, not significant.

Data Fig. 10b). In both contexts, very small changes to the amino acid sequence of the VSG resulted in large drops in binding (50–75%). We modelled the structure of these mosaics using ColabFold and their general structures matched AnTat1.1 (Fig. 5d), except at the disordered top of the N-terminal lobe. We found that mutations with the largest

effect on antibody binding are predicted to be at the apex of the structure whereas others can be found on the side of the monomer (Fig. 5e). These results demonstrate that, although the insertions characteristic of mosaic recombination are quite short, even these small changes to the VSG can confer large consequences for host antibody binding.

## Discussion

Chronic *T. brucei* infection relies on the generation of new VSGs, which dominate later stages of infection[1,2,4]. The inherent complexity of chronic infection, however, has obscured the underlying biological principles driving VSG diversification. Here we demonstrate that VSG diversification can be induced in vitro using Cas9-mediated double-strand DNA breaks within the VSG coding sequence, reproducibly generating mosaic VSGs that faithfully recapitulate mosaics formed naturally in vivo. By selecting just one VSG and looking at thousands of recombination outcomes, we have defined patterns characteristic of mosaic recombination. Mosaic VSGs typically form through short, templated insertions, and homology drives this process, restricting donor VSGs within the N terminus to those from a set of closely related family members. Using our in vitro system, we clarified the mechanisms underlying these patterns, showing that break-induced mosaic formation occurs via a templated gene conversion event that is dependent on both RAD51 and BRCA2, requires only 100 bp of imperfect sequence homology and can use suitable donor sequences regardless of their genomic position. Finally, we demonstrate that mosaic VSGs provide substantial immune evasion, particularly when these changes occur at the top of the VSG N terminus, which may reflect a hypervariable region within the VSG.

We find that VSG diversification and the majority of VSG switching by gene conversion are catalysed by the same machinery. Many VSG switch events are mediated by canonical homologous recombination, which relies on RAD51, an enzyme that facilitates recombination by binding single-stranded DNA ends at double-strand breaks and catalysing their insertion into a homologous DNA duplex. RAD51-dependent reactions in *T. brucei* are most efficient with stretches of homology of at least 100 bp, with mismatches markedly reducing recombination[37]. Our data suggest that mosaic recombination events exploit a form of RAD51-directed homologous recombination, with a short 100-bp length requirement, but appear to tolerate very high levels of base mismatch (about 20%) within the donor sequence. Although the precise nature of such a reaction is unclear, it is possible that the *T. brucei* mismatch repair machinery, which typically rejects longer donor sequences during homologous recombination when too many mismatches are present[38], is inactive or more relaxed during mosaic VSG formation. Although key aspects of the mechanism remain to be elucidated, our results indicate that *T. brucei* uses a particularly flexible form of RAD51-dependent homologous recombination to form mosaic VSGs. This error-tolerant mechanism allows even small, extremely degenerate VSG fragments to be used by the parasite for DNA repair, resurrecting ancient VSG sequences for use in novel antigen formation.

Notably, our analyses suggest that mosaic formation is the repair pathway of choice after a DNA break, at least within the VSG coding sequence, as we detect almost no switching after break induction. Additionally, the conserved 70-bp repeats interspersed among VSGs[7,8,28,29,39], which are thought to be critical mediators of the majority of VSG switch events by providing upstream homology to VSGs within the silent subtelomeric and minichromosomal repertoires[40], do not appear to have a role in mosaic formation, as an ectopic donor lacking these features can be used regardless of its genomic position. Therefore, the 70-bp repeats are probably not driving the initial homology search following a break within a VSG and the structure of the VSG archive has a much less prominent role in mosaic formation than we anticipated. Instead, it appears that the RAD51 machinery ultimately samples the entire genome for a donor that matches the DNA flanking the break, and in the absence of flanking homology, the homology search mechanism shifts to the larger genomic context of the double-strand break. In support of this, another recent study found that VSG recombinants arise more quickly after a DNA break than VSG switchers, suggesting significant attempts to repair the VSG are made before switching[24]. Determining

the timing of this transition from mosaic formation to VSG switching will be important to our understanding of VSG repair dynamics. Ultimately, the parasite DNA repair machinery appears to emphasize direct repair and the possibility of generating VSG diversity over a full VSG switch and the potential for immediate, complete immune evasion.

Although many mosaic events may not provide full immune escape, our data suggest that the parasite's recombination mechanism may tune outcomes to prioritize immune evasion. We found that the centre portion of the VSG gene appears to be especially successful at generating recombinants after a break; this region encodes the top of the N-terminal lobe of the VSG, a site directly exposed to host antibody and previously shown to be highly variable in human infections[10]. For AnTat1.1, and indeed most crystalized VSGs, this appears to be an unstructured region despite a variety of underlying structures[41]. We propose that this disordered region may have evolved to be more tolerant to amino acid changes, allowing diversification within the part of the protein most likely to facilitate immune evasion. Although many recombination events may occur after a DNA break, we hypothesize that only a subset, particularly those within disordered regions, maintain VSG structural integrity and allow the parasite to survive[42], resulting in an apparent increase in recombination within the regions of the VSG where mutation is most tolerated.

Indeed, even very small changes within the top of the VSG N-terminal lobe can confer substantial immune evasion, and mounting evidence[43,44] suggests that the host antibody response primarily targets this part of the VSG. Just a handful of amino acid substitutions can block more than 50% of antibody binding in polyclonal serum collected during experimental mouse infections. However, a 50% drop in binding may not be sufficient to fully evade all preexisting anti-VSG antibody, and the mosaic VSGs detected during chronic infections are typically complex. A wider diversity of AnTat1.1-derived mosaics can be observed within the tissues, however, where AnTat1.1-expressing parasites linger owing to differences in immune pressure[9]. We thus hypothesize that generating new VSGs is an iterative process, in which a series of recombination events progressively shifts the character of a VSG until a fully evasive variant is formed. The relatively 'protected' tissue spaces may serve as a site for this iterative process to occur. Another intriguing possibility is that partial immune evasion is advantageous for the parasite: a recent study suggested that exposure to sublethal antibody concentrations can trigger a VSG switch in *T. brucei*[45]. Perhaps parasites expressing these partially evasive variants are more prone to switching to a completely new VSG.

In addition to shedding light on the selective pressures imposed upon mosaic variants, our in vivo data also demonstrate that our in vitro model of antigenic diversification recapitulates VSG diversification in vivo. The recombination events we detect in μMT mice represent the full breadth of possible recombination events that AnTat1.1 can undergo, and these are largely represented within our in vitro break-induced mosaic VSG populations. However, the dominance of the 694 break position we observe in vitro is not reflected in our in vivo analysis. This could be for a variety of reasons, including non-uniform break locations along the VSG in vivo or a variety of DNA break types driving a different pattern of outcomes. Intriguingly, in the absence of an exogenous DNA break, we do not observe any diversification in vitro, whereas diversification occurs continuously in the μMT context. This suggests that some aspect of the host environment other than antibody pressure induces diversification in vivo. It remains to be investigated whether an in vivo cue for diversification exists.

The diversification mechanisms we have described, in which homology-templated DNA repair drives antigen evolution, may also be at play in other gene families for which diversity is critical. Many pathogens rely on diverse antigen genes to escape the host immune system, expressing one variable antigen from a silent repertoire stored within the damage-prone subtelomeres. Moreover, many eukaryotic pathogens that use

antigenic variation (for example, *Plasmodium*, *Giardia* and *Trypanosoma*) have lost canonical non-homologous end-joining DNA repair[46], suggesting an important role for alternative diversity-generating repair mechanisms. Notably, mosaic antigens have been observed in *Plasmodium falciparum* var genes[47–50] and *Giardia* variant-specific surface proteins[51,52], some of which resemble the small insertion mosaics we have described here. Given these observations, we suspect the repair pathway underlying mosaic VSG formation may contribute to the evolution of antigen repertoires in many diverse organisms. It is also plausible that this repair pathway facilitates the diversification of large gene families such as olfactory receptors[53,54] and protocadherins[53], which depend upon diversity and are known to swap sequences through gene conversion events[55–58]. A rigorous targeted approach, as we have used here, may be required to define the mechanisms underlying diversification of these other gene families.

Here we answer long-standing questions about how *T. brucei* generates new antigens, which are critical for maintaining a chronic infection. Using an in vitro toolkit for studying VSG diversification, we have defined some of the key molecular requirements underlying the formation of mosaic VSGs. More broadly, this study provides an experimental framework for the hypothesis-driven exploration of antigen diversification, not only in *T. brucei* but also in other pathogenic microorganisms.

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

## Methods

### Materials availability

New plasmids associated with this study were deposited to Addgene under Mugnier Lab and McCulloch Lab and are associated with this publication.

### Statistics and reproducibility

Sample size was not predetermined and experiments were not randomized or blinded. In vitro experiments were derived from two independently generated biological replicates, arising from different parentals except for the truncation experiments (same parental, but biologically unique clones), and offered high congruence sufficient for analysis. Mouse experimental size was selected based upon previous experiments with sufficient effect size observed[1,9]. WT versus μMT experiments were performed three independent times with independent vials of *T. brucei* stock using five mice of each genotype at the start of the infection. ΔRAD51 and Tb427VSG-8 infections in μMT mice were performed once from the same vial of stock, infecting four and five mice, respectively. Flow cytometry experiments were performed twice for Fig. 5b, with one sample, mosaic 5, repeated across both experiments, and once for Fig. 5c. Data presented are technical replicates. Antisera collected from mice represent four biological replicates.

A Shapiro–Wilk normality test was performed for all statistics calculated from mouse experiments, followed by the specified Wilcoxon test. Significance from FACS experiments was determined from a one-way ANOVA with post hoc Tukey HSD. All graphs include mean if $n = 2$ or mean ± s.d. if $n \geq 3$. Calculations were performed in R (v.4.0.2). Details with exact $P$ values and statistical test results can be found in Supplementary Data 2.

### Parasites

Pleiomorphic EATRO1125 AnTat1.1 90-13 *T. brucei* parasites were maintained in HMI-9 media with 10% heat-inactivated FBS (F0960, 500 ml, Sigma-Aldrich) and 10% Serum Plus (14008C, 500 ml, Sigma-Aldrich) or in HMI-11 with 10% FBS where specified[60]. These parasites originated from K. Matthews. Parasites were passaged when they reached approximately $5 \times 10^5$ cells per ml unless otherwise specified. Monomorphic Single Marker Lister427 VSG221 TetR T7RNAP bloodstream form (NR42011; LOT: 61775530)[61] *T. brucei* were maintained in HMI-9 up to $1 \times 10^6$ parasites per ml. VSG221 has since been renamed to VSG-2. Monomorphic Single Marker 427 1339 Cas9 TetR T7RNAP (bloodstream form) (NR-56793; LOT: 70056027) *T. brucei* subsp. *brucei* were maintained in HMI-9 up to $1 \times 10^6$ parasites per ml. This was obtained through BEI Resources, National Institute of Allergy and Infectious Diseases (NIAID), National Institutes of Health (NIH). EATRO1125 AnTat1.1 J1339 pleiomorphic parasites were a gift from K. Matthews[62]. Parasites were verified for expected VSG expression via amplicon sequencing. Parasite cultures were not tested for mycoplasma contamination.

### Plasmids

Plasmids were synthesized with Gibson Assembly with a custom master mix[63] unless otherwise specified. Whole plasmid sequencing was performed by Plasmidsaurus using Oxford Nanopore Technology and their custom analysis and annotation pipelines. Detailed description of plasmid construction, using published publicly available plasmids[64–69], can be found in the Supplementary Methods.

### Transgenic parasites

To obtain transgenic parasites, 5–20 million parasites were electroporated with 5–10 μg of digested plasmid with an AMAXA Nucleofector II using X-001 in Human T-cell Nucleofector Solution (Lonza, VPA-1002) or using Z-001 in Tb transfection buffer[70]. All parasites were maintained in selection unless otherwise specified. Detailed descriptions[64,71–74]

of the generation of transgenic parasites can be found in the Supplementary Methods.

### T7-guide synthesis and purification

Guide RNAs were designed with EuPaGDT[75]. DNA fragments containing a T7 promoter and guide RNA sequence were synthesized as described previously[76]. PCR products from 12 identical PCRs were pooled and purified via ethanol precipitation.

### Cas9 transient electroporations

Approximately 24 h before electroporation, parasites were seeded at a density of 83,000 parasites per ml and induced with 1 μg ml⁻¹ doxycycline (Millipore Sigma, D9891-1G). The total culture volume was determined by the number of samples being electroporated; 12 ml was seeded per sample. Blasticidin selection (Cas9) and puromycin selection (silent VSG-228) were maintained, if applicable, and for Lister427 parasites expressing AnTat1.1, hygromycin (active VSG expression) was removed at this stage. Either 8 ml (Lister427) or 10 ml (EATRO1125) of the bulk parasite culture, approximately 5–10 million cells, was spun down for each sample. Medium was then removed, and parasites were resuspended in 100 μl of Human T-cell Nucleofector Solution. Each sample was electroporated using the X-001 program on the AMAXA Nucleofector II with approximately 1–1.5 μg of purified T7-guide in a volume less than 10 μl or a sample without any DNA as a negative control. Parasites were moved into 5 ml of HMI-9 in six-well plates to recover for 30 min then moved into 20-ml total in flasks to recover overnight. About 24 h after electroporation, parasites were counted and split. For EATRO1125, 12 million cells (or all cells if there were fewer than 12 million) were seeded into 60-ml total with blasticidin. For Lister427, 2 million cells were seeded into 20-ml total with blasticidin. At the 48-h time point, parasites were counted, collected and stored in TRIzol (Invitrogen, 15596026) for subsequent RNA extraction.

For the constitutively expressed Cas9 parasites, 1.7–2.3 μg of guide was electroporated into 5–10 million cells. Parasites were grown in 20–30 ml of HMI-11 media with 0.2 μg ml⁻¹ puromycin for 24 h to recover, counted, then split into 60 ml of HMI-11 media with puromycin for the final 24 h. Twelve million cells (or all cells if there were fewer than 12 million) were seeded into 60-ml total. At the 48-h time point, parasites were counted, collected and stored in 15 μl of PBS with 150 μl of RNAlater (Invitrogen, AM7020) at −20 °C for subsequent RNA extraction.

### Isolation of mosaic-expressing parasites

Parasites were isolated from inducible Cas9-expressing parasites. A plasmid expressing an sgRNA (either pT7-sgRNA or pLEW-T7-sgRNA) was inserted into each parental clone and then Cas9 was induced with doxycycline. Multiple dilutions of parasites were plated in 96-well plates and selected from plates with fewer than 30 surviving parasite clones. Upon isolation of a mosaic parasite clone, all drugs were removed.

EATRO1125 parasite clones expressing mosaic AnTat1.1 were isolated from parental lines with both pT7-sgRNA (guides 243 or 694; drugs: 5 μg ml⁻¹ blasticidin and 1 μg ml⁻¹ doxycycline) and pLEW-T7-sgRNA (guides 243, 694 or 1459; drugs: 5 μg ml⁻¹ blasticidin, 0.1 μg ml⁻¹ puromycin and 1 μg ml⁻¹ doxycycline).

EATRO1125 parasite clones expressing mosaic Tb427VSG-8 were isolated from parental lines with pLEW-T7-sgRNA (guide 783; drugs: 5 μg ml⁻¹ blasticidin, 0.1 μg ml⁻¹ puromycin and 1 μg ml⁻¹ doxycycline). EATRO1125 parasite clones expressing mosaic EATRO1125VSG-73 were isolated from parental lines with pLEW-T7-sgRNA (guides 194, 680 and 1436; drugs: 5 μg ml⁻¹ blasticidin, 0.1 μg ml⁻¹ puromycin and 1 μg ml⁻¹ doxycycline).

### VSG PCR and identification via sequencing

VSG sequences for clones were determined from extracted RNA. Complementary DNA was synthesized using the Superscript III

Reverse Transcriptase (Invitrogen, 18080051) and a VSG-specific primer that binds to a conserved 14-bp sequence within the 3′ UTR (5′-GTGTTAAAATATATC-3′). Then, 2 µl of RNase-treated cDNA was amplified for 35 cycles with VSG-specific primers: a spliced-leader (5′-ACAGTTTCTGTACTATATTG-3′) and SP6-VSG 14-mer (5′-GATTTAGG TGACACTATAGTGTTAAAATATATC-3′) using Phusion polymerase (Thermo Fisher, F530L) (annealing temperature 55 °C, extension 45 s). PCR products were cleaned using the Monarch PCR & DNA cleanup kit (NEB, T1030L). VSG sequences were determined by amplicon nanopore sequencing performed by Plasmidsaurus using Oxford Nanopore Technology with their custom analysis and annotation in which fragmented VSG amplicons were sequenced and assembled into full-length sequences or targeted Sanger sequencing with the sequencing primer (5′-AGAGAATACTAAGCTAGTTGGC-3′) performed by Azenta Life Sciences.

## Assessment of donor VSG following mosaic formation

Lister427 parasites containing constitutively expressed Cas9, active VSG AnTat1.1, a silent inserted VSG-228 donor and constitutively expressed guide targeting AnTat1.1 position 694 were expressing mosaic AnTat1.1 by the time they recovered from guide insertion, but only 1/12 positive colonies were clonal as determined by Sanger sequencing of the expressed VSG (5′-AGAGAATACTAAGCTAGTTGGC-3′) performed by Azenta Life Sciences. The VSG was amplified as detailed above in 'VSG PCR and identification via sequencing'. The resulting sequencing traces were visually inspected to determine candidates likely to have a high concentration of cells with VSG-228 donation. These were subcloned and individual clones were obtained, with the analysis repeated to determine donor VSG. gDNA was isolated from cells with a VSG-228 donation. In brief, parasites were collected via centrifugation at 2,600$g$ for 4 min. Parasites were washed once with around 500 µl of PBS, spun at 2,600$g$ for 4 min and the PBS removed. DNA was extracted using the Monarch Spin gDNA Extraction Kit (NEB, T3010S). Approximately 40 ng of gDNA was used to amplify the donor VSG cassette, VSG-228, with one primer within the plasmid backbone and the other within the Blasticidin resistance gene (40 ng of input gDNA; Fwd: 5′-TTGACACCAGTGAAG ATGCGG-3′; Rev: 5′-CGGCAGTTTACGAGAGAGATGA-3′; annealing temperature 60 °C, extension 90 s) for 35 cycles using Phusion DNA Polymerase (NEB, M0480L). The full sequence of the amplicon was determined by Plasmidsaurus using Oxford Nanopore Technology with their custom analysis and annotation. These were BLASTed (NCBI)[77] against the original plasmid sequence.

## Mouse infections

All experiments involving mice were performed in accordance with the protocol approved by the Institutional Animal Care and Use Committee at Johns Hopkins University. Mice were housed at 68–76 °C with 30–70% relative humidity (target 42%) under a 14.5 h:9.5 h light:dark photoperiod. The 8–12-week-old female C57BL/6 mice and µMT⁻ (B6.129S2-Ighm^tm1Cgn/J)[35] mice (Jackson Labs) were infected with about five EATRO1125 parasites by intravenous injection in the tail vein. These parasites express either AnTat1.1 or VSG-421. Blood was collected 6 days after infection through a submandibular bleed. At 15 days after infection, mouse blood was collected by cardiac puncture.

The 8–12-week-old female µMT⁻ mice were infected with about five Tb427VSG-8 EATRO1125 inducible Cas9 parasites. Blood was collected 6 days after infection through a submandibular bleed. At 13–15 days after infection, mouse blood was collected through a submandibular bleed. For one mouse, further blood was collected by cardiac puncture 18 days after infection.

The 8–12-week-old female µMT⁻ mice were infected with about ten EATRO1125 ΔRAD51 parasites. Blood was collected 7–8 days after infection through a submandibular bleed. At 14 days after infection, mouse blood was collected by cardiac puncture.

Starting 4 days after infection, parasitaemia was monitored within mice every 2 days via tail bleed. After blood collection, extra gel packs and in-cage food pellets were provided to mice during recovery. Blood was stored in TRIzol LS for RNA extraction (Invitrogen, 10296028).

## Tb427VSG-8/EATRO1125VSG-73 isolation

Tb427VSG-8-expressing EATRO1125 parasites were obtained by infecting a C57BL/6 mouse with five inducible Cas9 EATRO1125 AnTat1.1-expressing parasites intravenously. Starting 4 days after infection, parasitaemia was monitored within mice every 2 days via tail bleed. At 16 days after infection, the second peak of parasitaemia where AnTat1.1-expressing parasites are undetectable in the blood, mice were euthanized, and blood was collected via cardiac puncture. Parasites were cultured for 4 days in the presence of 5 µg ml⁻¹ blasticidin and 1:100 Penicillin-Streptomycin (Gibco, 15140122) then frozen in HMI-9 with 10% glycerol. Parasites were thawed and subcloned into 96-well plates. Positive colonies were pulled from plates with fewer than ten colonies. Parasites were maintained in 5 µg ml⁻¹ blasticidin. These parasites double roughly once per day.

EATRO1125VSG-73-expressing parasites were obtained by inducing Cas9 with 1.0 µg ml⁻¹ doxycycline in EATRO1125 AnTat1.1-expressing parasites for 24 h. Parasites were then electroporated with approximately 2 µg of guide targeting the sequence upstream of AnTat1.1 (AnTat1.1 upstream target: 5′-CAAAAAGGAGGAGAGGAAAT-3′). These were cultured for 7 days in the presence of 1:1,000 rabbit anti-AnTat1.1 antibody (J. Bangs), 5 µg ml⁻¹ blasticidin and 1.0 µg ml⁻¹ doxycycline and then subcloned into 96-well plates. The clone used was pulled from a plate with fewer than ten colonies.

After obtaining single colonies with a novel VSG, VSG sequences were determined as described in 'VSG PCR and identification via sequencing'.

## RNA preparation

Parasites were stored in TRIzol before RNA extraction. Blood with parasites was stored in TRIzol LS. RNA was extracted via phenol/chloroform extraction according to the manufacturer's protocol. Purified RNA was DNase treated with Turbo DNase (Thermo Fisher, AM2239) and purified with 1.8X Mag-Bind TotalPure NGS Beads (Omega Bio-tek, M1378-01). Verification of effective DNase treatment was performed via PCR of hygromycin (EATRO1125 only) (Fwd: 5′-ACAGCGGTCATTGACTGGAG-3′; Rev: 5′-ATTTGTGTACGCCCGACAGT-3′, annealing temperature 52 °C, extension 30 s) or HSP70 (Lister427 & EATRO1125) (Fwd: 5′-AGAACACT ATCAATGACCCCAAC-3′; Rev: 5′-CCATGCCCTGGTACATCT-3′, annealing temperature 50 °C, extension 15 s) genes for 30 cycles using OneTaq DNA Polymerase.

For the EATRO1125 RAD51 and BRCA2 knockout experiments, parasites were stored in RNAlater and shipped on dry ice. To extract RNA, parasites were spun at 2.6$g$ for 4 min at 4 °C. RNAlater was removed and 1 ml of TRIzol was added. Some samples did not have a tight cell pellet and thus were spun multiple times and the RNAlater supernatant was carefully removed incrementally following each spin. For some samples, the RNAlater could not be removed entirely. These samples were extracted twice with TRIzol. The RNA pellet mixed with remaining RNAlater at the end of the isopropanol precipitation was resuspended in 1 ml of TRIzol and the extraction was repeated for clean RNA isolation.

## VSG-seq

VSG-seq was performed as previously described[1,9]. Libraries were sequenced with 100-bp single-end reads on a NovaSeq6000. Analysis was performed using the VSGSeqPipeline found at github.com/mugnierlab.

## VSG-AMP-seq library preparation and analysis

VSG-AMP-seq was based upon anchored multiplex PCR (AMP)-seq[78] and genome-wide, unbiased identification of DNA double-stranded breaks enabled by sequencing (GUIDE-seq)[59]. Detailed descriptions of

the library prep and analysis pipeline, including all software dependencies[79–81], can be found in the Supplementary Methods.

**Identification of mosaic VSG from ORFs.** ORFs from ref. 9 were tiled into 20-bp *k*-mers overlapping by 1 bp each. All tiles were aligned to AnTat1.1 using bowtie[82] with the following flags: -v 2. ORFs in which 15 or more tiles successfully aligned to the AnTat1.1 sequence were identified as possible mosaics (≥35-bp match, potentially discontinuous). AnTat1.1 sequences containing ORFs were assessed for sequence matching common donor VSGs: VSG-228, VSG-2986, VSG-3110 and VSG-7358. If a best match could not be determined or if recombination did not occur with one of the typical donor VSGs, VSG sequences were analysed by hand to identify remaining mosaic recombination events.

**VSG clustering and family identification.** VSGs were identified from published *T. brucei* genomes[6,27,83,84] and expressed VSGs[9]. VSGs were clustered into family groups using either BLASTn[77] or UCLUST algorithms[85]. Further details, including further software dependencies[86,87], can be found in the Supplementary Methods.

## Western blotting
EATRO1125 parasites with pLEW-FLAG-La-Cas9 and pLEW-T7sgRNA were grown for 24 h in the presence of 1 µg ml⁻¹ doxycycline or DMSO vehicle control. After induction, 5 million parasites were collected and washed with 25 °C PBS. Then, pelleted parasites were resuspended in 50 µl of RIPA buffer (50 mM Tris, 150 mM NaCl, 1% NP-40, 0.5% sodium deoxycholate, 0.1% SDS, pH 7.4) + 2 × Laemmli buffer. Lysed parasites were boiled at 95 °C for 5 min. Then, 5 µl of lysates were separated on a Tris-glycine polyacrylamide gel at 110 V for 100 min in Tris/glycine running buffer (25 mM Tris, 192 mM glycine, 0.1% SDS, pH 8.3). Proteins were then transferred onto a nitrocellulose membrane using transfer buffer (25 mM Tris, 192 mM glycine, 20% methanol) overnight at a 60-mA per transfer box at 25 °C. For FLAG (1:1,000) and EF1α (1:1,000), lysates were separated on 10% polyacrylamide gels and transferred onto 0.45-µm membranes. For γ-H2A (1:200), lysates were separated on a 15% polyacrylamide gel and transferred onto a 0.2-µm membrane. After transfer, blots were blocked with 0.5% BSA in TBS with 0.05% Tween 20 (TBST, 20 mM Tris, 150 mM NaCl, pH 7.6) for an hour at room temperature under constant agitation. Blots were incubated with primary antibody (see above dilutions) for 2 h at 25 °C. After five TBST washes, blots were stained for 1 h with goat anti-mouse (1:5,000, Cell Signaling, 7076S) or goat anti-rabbit-HRP-conjugated secondary (1:5,000, Cell Signaling, 7074S). After another five washes, blots were incubated with ECL (Cytiva, RPN2109) and film was exposed to blots in a dark room. Developed film was scanned and images were processed with FIJI (v.2.14.0/1.54.f)[88]. Primary antibodies used were mouse anti-FLAG (M2 clone) (Millipore Sigma, F3165-1MG), mouse anti-EF1α (CBP-KK1 clone) (Millipore Sigma, 05-235) and rabbit anti-γ-H2A, a kind gift from G. Hovel-Miner based upon ref. 89. Full uncropped images can be found in Supplementary Fig. 3.

## Death assays and clonal outcome analysis
Multiple parasite clones detailed in 'Isolation of mosaic-expressing parasites' using the pLEW-T7-sgRNA with guides at cut sites 243, 694 or 1459 were replica plated in 96-well plates in 5 µg ml⁻¹ blasticidin, 0.1 µg ml⁻¹ puromycin and 1 µg ml⁻¹ doxycycline or an equal volume of DMSO at multiple parasite dilutions. Uninduced replica plates had at most 76 colonies. Induced replica plates had at most 13 colonies. Individual clones were counted after 14 days. Upon isolation of a parasite clone, all drugs were removed. A subset of parasites were analysed to determine the VSG expressed.

VSG sequences for clones were determined as above in 'VSG PCR and identification via sequencing'. Consensus.ab1 sequencing files were assessed for their clonality (Clonal: high agreement at all bases, versus Polyclonal Colony: disagreement at multiple base locations).

## Tb427VSG-8 mosaic detection in vivo
RNA from day 6, day 13, day 15 and day 18 mouse blood was extracted and DNase treated as above. cDNA was synthesized using the Superscript III Reverse Transcriptase and a VSG-specific primer that binds to a conserved 14-bp sequence within the 3′ UTR (5′-GTGTTAAAATATATC-3′). cDNA was treated with Rnase A and RNase H for 30 min then purified with 1.8X Mag-Bind Total NGS Beads. Purified cDNA was amplified for 25 cycles with VSG-specific primers: a spliced-leader (5′-ACAGTTTCTGTACTATATTG-3′) and a sample-specific barcoded SP6-VSG 14-mer (or one without the barcode, see VSG barcoding primers, Supplementary Table 1), using Phusion polymerase (annealing temperature 55 °C, extension 45 s). This PCR product was purified with a 1.8X bead cleanup.

Samples were pooled, prepared with Ligation sequencing DNA V14 kit (SQK-LSK114) then sequenced on an Oxford Nanopore Technology PromethION with a 10.4.1 flow cell yielding approximately 100.29 Gb of data with an average length of 1.67 kb.

## Tb427VSG-8 and EATRO1125VSG-73 in vitro and in vivo mosaic identification and analysis.
Identification of mosaic VSGs from mixed in vitro and in vivo samples, including software dependencies[77,79,90–93], is detailed in the Supplementary Methods.

## Mouse AnTat1.1 antiserum generation
Four 8–12-week-old female C57BL/6 mice were infected with about 100 Lister427 SM-AnTat1.1 transgenic parasites via intravenous injection in the tail vein. Parasites appeared in mice 4–5 days after infection. On day 5 after infection, mice were treated with 2 µg of Berenil (Cayman Chemical, 18678) in PBS via intraperitoneal injection to cure infection. A second dose of Berenil was repeated 24 h later. At 14 days after infection, mice were humanely euthanized and blood was collected via cardiac puncture. Serum was isolated using microtainer gel tubes (BD, 365967), spun at 6,000*g* for 3 min and stored at −80 °C until use.

## Flow cytometry
In 96-well plates, 200,000 parasites were stained with 1:20,000 rabbit anti-AnTat1.1 primary antibody[36] (J. Bangs) or 1:1,000 mouse anti-AnTat1.1 serum for 10 min at 4 °C while shaking in PBS + 10 mg ml⁻¹ glucose. Parasites were washed once with 100 µl of PBS + glucose. Then, parasites were stained with Alexa Fluor 647-conjugated goat anti-rabbit IgG (H+L), F(ab')2 Fragment (Cell Signaling, 4414S) or goat anti-mouse IgG (H+L), F(ab')2 Fragment (Cell Signaling, 4410S), respectively, at 1:1,000 at 4 °C while shaking in PBS + glucose. After washing again with 100 µl of PBS + glucose, parasites were resuspended in PBS + glucose + 1:20 propidium iodide (BD Biosciences, 556463) and analysed on an Attune Nxt flow cytometer (Invitrogen). Data analysis was performed using FlowJo (v.10.8.1). A gating strategy is provided in Supplementary Fig. 2.

## Analysis and modelling of VSG N-terminal domains
Full-length protein coding sequences from AnTat1.1 and its isolated mosaics were used for structural modelling. Only N-terminal domain sequences were used for protein structural prediction as this region of the VSG is the most well-defined experimentally. Signal peptides are cleaved from the mature VSG during processing, so we used SignalP 6.0 (ref. 94) to predict and remove the Sec/SPI sequence (--organism eukarya, --mode fast), resulting in a FASTA file of mature proteins. To determine the coordinate of the N-terminal domain, we used a Python analysis pipeline available at https://github.com/mugnierlab/find_VSG_Ndomains. The script identifies

the boundaries of the VSG N-terminal domain using the HMMscan function under HMMer v.3.1b2 (ref. 95). Query sequences are searched against an HMM profile containing 735 known N-terminal domain sequences from ref. 6 and N-terminal domains defined by the largest envelope domain coordinate that meets the $E$ value threshold ($1 \times 10^{-5}$, -domE 0.00001). The processed FASTA file containing only mature VSG N-terminal domain sequences was used as input for structural prediction of monomers using LocalColabFold (v.1.5.5)[96] function colabfold_batch, run using the following arguments: --amber, --templates, --num_recycle 3. The best-ranked output model with the highest average predicted local distance test score (pLDDT), that is, the highest-confidence model, was used for downstream analyses. Model visualization and alignment were performed using UCSF ChimeraX v.1.7.1 (ref. 97), developed by the Resource for Biocomputing, Visualization, and Informatics at the University of California, San Francisco, with support from the NIH (grant no. R01-GM129325) and the Office of Cyber Infrastructure and Computational Biology, NIAID.

## Reporting summary

Further information on research design is available in the Nature Portfolio Reporting Summary linked to this article.

## Data availability

Raw sequencing reads from VSG-AMP-seq, VSG-seq and nanopore Tb427VSG-8 amplicons were deposited to the NCBI Sequence Read Archive under project no. PRJNA1140873. Further data used in this study are detailed in the Supplementary Methods[6,9,27,83,84]. Source data are provided with this paper.

## Code availability

Original code associated with this study is available from GitHub at github.com/mugnierlab/Smith2026 and from Zenodo at https://doi.org/10.5281/ZENODO.18716076 (ref. 98). VSG-AMP-seq is available from GitHub at github.com/mugnierlab/Smith2026/tree/main/VSG-AMP-Seq and from CodeOcean at https://doi.org/10.24433/CO.3005244.v1 and https://doi.org/10.24433/CO.3283350.v1.

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

**Acknowledgements** We thank J. Bangs for generating and sending us primary rabbit anti-AnTat1.1 polyclonal antibody. Thanks to G. Hovel-Miner for the primary rabbit anti-γ-H2A antibody. We appreciate members of the Mugnier laboratory for their thoughtful review of this manuscript, particularly C. Duque and S. Gutierrez. We thank D. Mohr and the JHU GCRF Genomics Core for sequencing services. Thanks to J. Verdi for pUC19-HYG-BES1-VSG-3-S317A-telo and advice about the generation of transgenic parasites. Data analysis was carried out at the Advanced Research Computing at Hopkins (ARCH) core facility (rockfish.jhu.edu), which is supported by the National Science Foundation (NSF) grant no. OAC 1920103. This work was supported by the JHU Catalyst Award. J.E.S., J.M.C.H., A.K.B., J.S., B.Z. and M.R.M. were supported by the Office of the Director, NIH (grant no. DP5OD023065). J.E.S. and J.S. were additionally supported by NIH training grant no. T32GM007 and J.E.S. was supported by the National Science Foundation (NSF) Graduate Research Fellowship under grant nos. 1232825 and 1746891. J.M.C.H., A.K.B., J.S. and E.H.A. were additionally supported by NIH grant no. T32AI007417. J.C.M. and R.M. were supported by Wellcome Trust grants no. 206815/Z/17/Z and no. 224501/Z/21/Z.

**Author contributions** J.E.S. and M.R.M. conceptualized the study. J.E.S., J.M.C.H., A.K.B., A.M., S.D.G.-S., B.Z. and E.H.A. were responsible for the methodology. J.E.S., A.M., J.S. and L.M. were responsible for the software. J.E.S., K.J.W., E.M.K. and J.C.M. performed the validation. J.E.S. performed the formal analysis. J.E.S., K.J.W., E.M.K., J.C.M., L.S., J.S., A.K.B., A.M., J.Z., D.N.M. and Z.E.B. performed the investigations. R.M. and M.R.M. were responsible for resources. J.E.S., A.M. and J.S. were responsible for the data curation. J.E.S. and M.R.M. wrote the original draft of the manuscript. J.E.S., A.K.B., R.M. and M.R.M. reviewed and edited the manuscript. J.E.S. and J.S. performed the visualization. J.E.S. and M.R.M. supervised the study. J.E.S. was responsible for project administration. J.E.S. and M.R.M. were responsible for funding acquisition. VSG-AMP-seq was developed by J.E.S., S.D.G.-S. and B.Z. EATRO1125 Cas9 experiments were performed by J.E.S. Lister427 Cas9 and Lister427-Cas9-AnTat1.1 experiments were performed by J.E.S. Lister427-Cas9-AnTat1.1 with VSG-228 experiments were performed

by K.J.W. Lister427-Cas9-AnTat1.1 with truncated VSG-228 experiments were performed by E.M.K. EATRO1125 Cas9 RAD51 and BRCA2 knockout experiments were performed by J.C.M. Molecular cloning was performed by J.E.S., K.J.W., E.M.K., J.C.M., D.N.M. and Z.E.B. Transgenic parasites were obtained by J.E.S., K.J.W., E.M.K. and J.C.M. Western blots were performed by J.E.S. and E.M.K. Parasite death assays were performed by E.M.K. and Z.E.B. The VSG-AMP-seq pipeline was written by J.E.S. RNA extractions, DNase treatment and VSG PCRs for sequencing were performed by J.E.S., K.J.W., L.S. and A.K.B. VSG-seq was performed by J.E.S. and A.K.B. VSG-AMP-seq was performed by J.E.S., K.J.W. and L.S. Mouse infections and all downstream analyses were performed by J.E.S. ORF mosaic recombination identification was performed by J.E.S. and A.M. Mosaic parasite clones were processed for sequencing by J.E.S., K.J.W., E.M.K., J.Z. and Z.E.B. FACS experiments and analyses were performed by J.E.S. Modelling of VSGs was performed by J.S. and L.M. The Lister427 2018 VSG annotation pipeline was written by J.E.S. VSG clustering was performed by J.E.S., J.S., A.M., J.M.C.H. and E.H.A. qPCR was performed by J.C.M.

**Competing interests** The authors declare no competing interests.

**Additional information**
**Correspondence and requests for materials** should be addressed to Monica R. Mugnier.

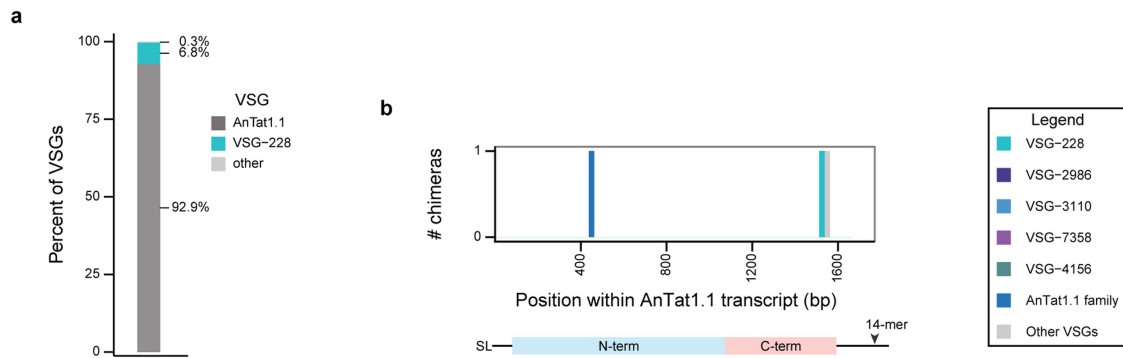

**Extended Data Fig. 1 | Mosaic Recombination Site & VSG-AMP-seq Validation.** A) A stacked bar graph quantifying the mix of Lister427 parasites engineered to express either AnTat1.1 or VSG-228 as determined by VSG-seq. These VSGs are naturally absent from Lister427 parasites and therefore chimeric reads containing sequence from both VSGs must have arisen from error. B) A histogram showing the midpoint of chimeric recombination events misidentified as a mosaic VSGs by VSG-AMP-seq from the mixed parasite sample in A.

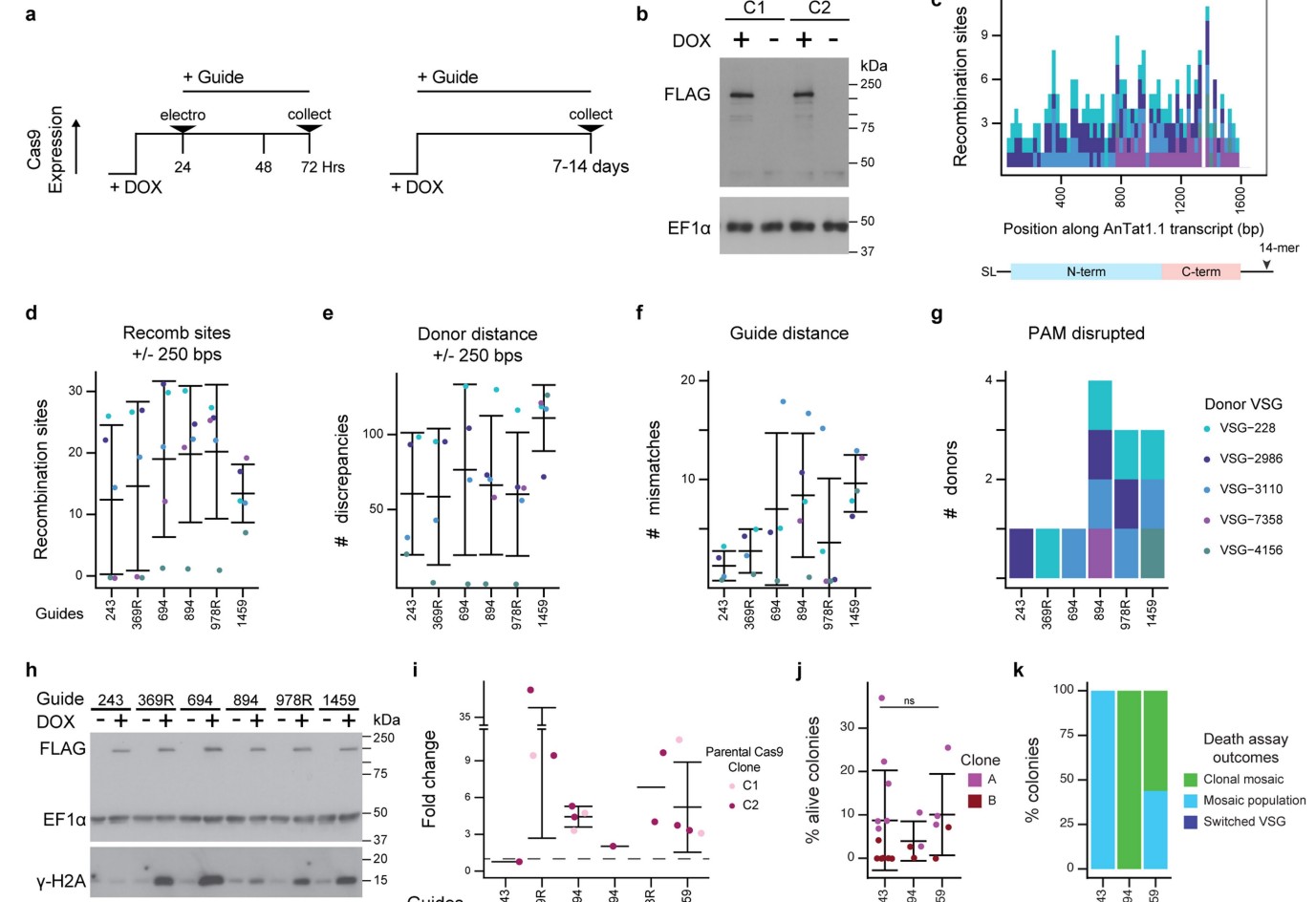

**Extended Data Fig. 2 | Cas9 system and sgRNA design and analysis.**
A) A schematic of the Cas9 induction experiment. Transient electroporation shown on the left while constitutive guide expression to isolate mosaic clones shown at right. B) An immunoblot showing FLAG-tagged Cas9 induction 24 h following doxycycline treatment (DOX). DMSO was used as a vehicle control. EF1α was used as a loading control. C) A histogram showing the midpoint of all possible recombination sites 5 bp or longer between AnTat1.1 and its family members. D) Quantification of the number of recombination sites within 250 bp up or downstream of the cut site for each donor VSG. Mean ± s.d. E) Quantification of the Levenshtein distance between AnTat1.1 and family members. Mean ± s.d. F) Quantification of the number of mismatches at the guide binding site between AnTat1.1 and family members. Mean ± s.d. G) Histogram of which donor VSGs can disrupt the PAM when used to repair

AnTat1.1. H) Representative immunoblot of FLAG, EF1α, and γ-H2A from doxycycline-induced clones and uninduced controls. I) Quantification of the induction of DNA damage, as measured by γ-H2A phosphorylation, following 24 h of induction. Mean ± s.d. n = 1–4, multiple independently-generated clones were induced. J) Quantification of colony survival after Cas9-induced cutting at positions 243, 694, or 1459 within AnTat1.1. Statistical significance was determined with a one-way ANOVA with post-hoc Tukey HSD. Mean ± s.d. n = 4–12. K) Quantification of the sequencing results for a subset of surviving parasite clones from J post Cas9-cutting. These clones were determined to be either a clonal mosaic VSG, a mixed population including one or more mosaic VSGs, or a switched VSG. The mixed populations also could include intact AnTat1.1. n = 6–14 SL = 5′ splice leader sequence, 14-mer = 3′ sequence conserved in all VSG transcripts.

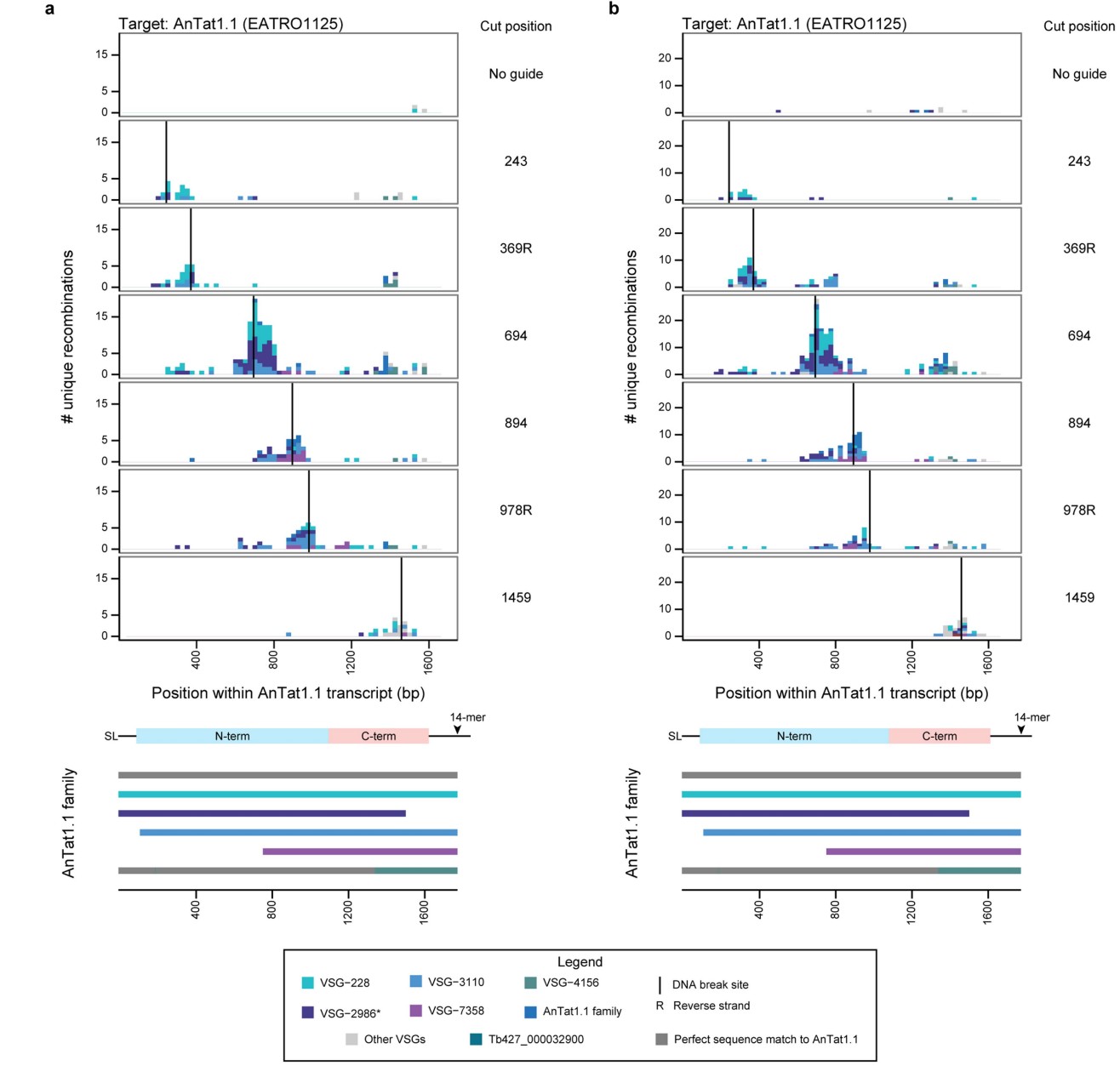

**Extended Data Fig. 3 | In vitro Cas9 induction extended data.** Histogram of unique recombination events identified along the AnTat1.1 transcript after induction of a double strand DNA break for two independently-generated clones, Clone 1 (A) and Clone 2 (B). The Cas9 DNA break site is indicated by a vertical line. Cut positions, relative to the 5′ end of the VSG transcript are: 243, 369, 694, 894, 978, and 1459. Sections of the histogram are colored to indicate donor VSG identified. R indicates a guide that binds to the reverse strand. The midpoint of the perfect homology between AnTat1.1 and the donor VSG at the recombination site is plotted. If a mosaic sequence matched >1 potential donor VSG, the average recombination position was plotted. Below histograms are schematics of the AnTat1.1 family aligned to the AnTat1.1 transcript. Gray sequences are a perfect match to AnTat1.1. Clone 2 data for positions 243, 694, and 1459, along with the negative control are also shown in Fig. 2a.

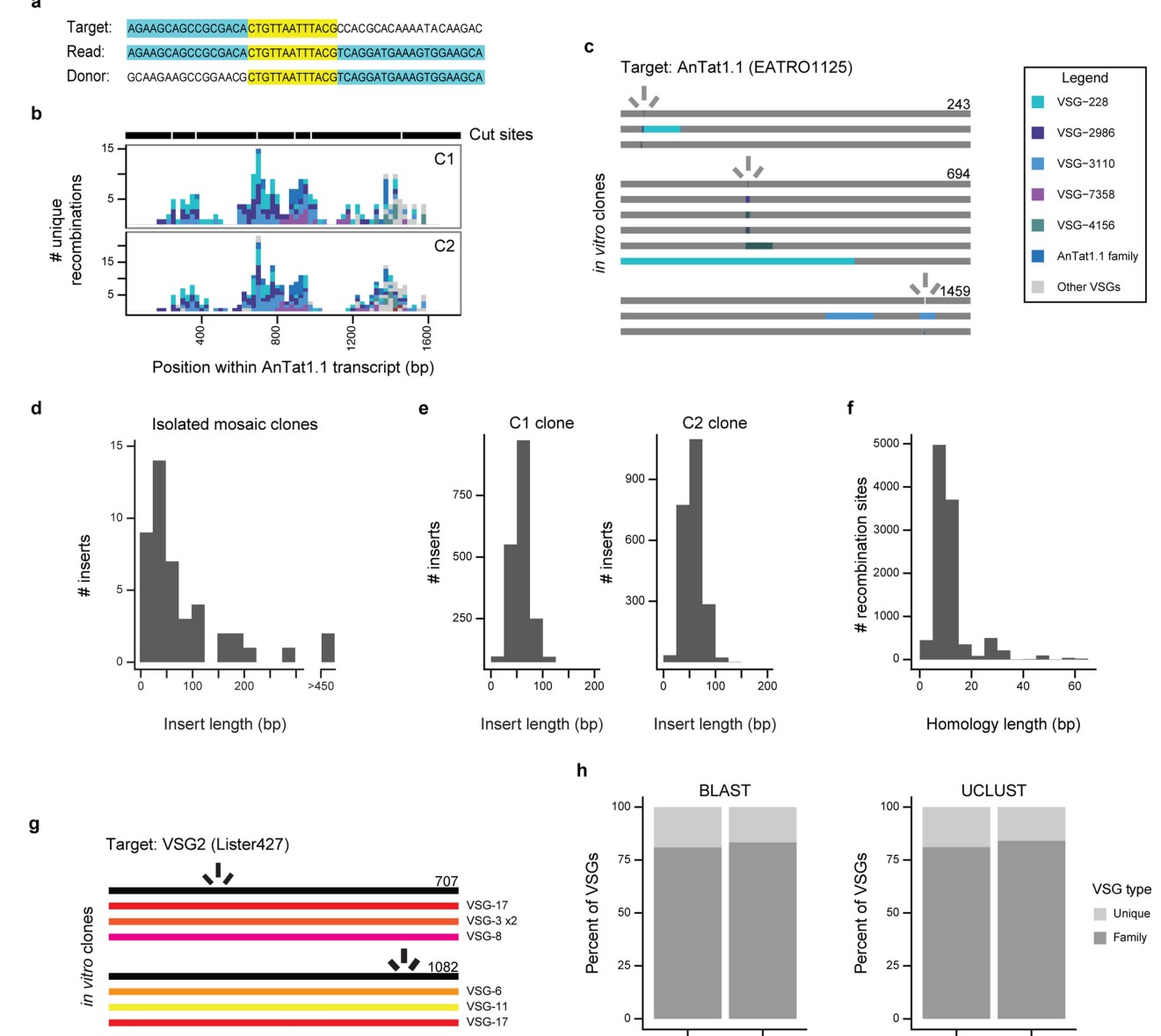

**Extended Data Fig. 4 | Characteristics of mosaic VSG recombination.**
A) A schematic of a recombination site. The region in yellow is the putative recombination site, shared by all 3 sequences. B) Histogram with a summary of the unique recombination sites found within guide-induced break for each clone. Cut sites are indicated above the histograms as gaps within the black line. The midpoint of the perfect homology between AnTat1.1 and the donor VSG at the recombination site is plotted. If a mosaic sequence matched >1 potential donor VSG, the average recombination position was plotted. The legend for the donor VSG colors is to the right. SL = 5′ splice leader sequence, 14-mer = 3′ sequence conserved in all VSG transcripts C) Representative schematics of mosaic VSGs from parasite clones isolated after DNA breaks within AnTat1.1. n = 46 D) A histogram of donor VSG insertion lengths identified in all isolated mosaic-expressing clones. The insert length only includes newly inserted

sequence and does not include recombination sites. E) Histograms of donor VSG insertion lengths identified in all mosaic VSG reads from Fig. 2a and Extended Data Fig. 3. The insert length only includes newly inserted sequence and does not include recombination sites. Clone C1 and C2 sequenced via VSG-AMP-seq are shown separately. The limit of detection for an insertion is approximately 200 bp. F) A histogram of the length of the shared identity between AnTat1.1 and the donor VSG at each recombination site. G) A schematic showing identified switchers following a Cas9-induced double-strand break within VSG-2 at locations 707 and 1082. All isolated clones shown. H) Quantification of types of VSGs within the VSGnomes from EATRO1125 and Lister427 parasites using BLAST (left) or using the UCLUST greedy clustering algorithm (right). Perfectly duplicated VSGs without another family member were counted as unique.

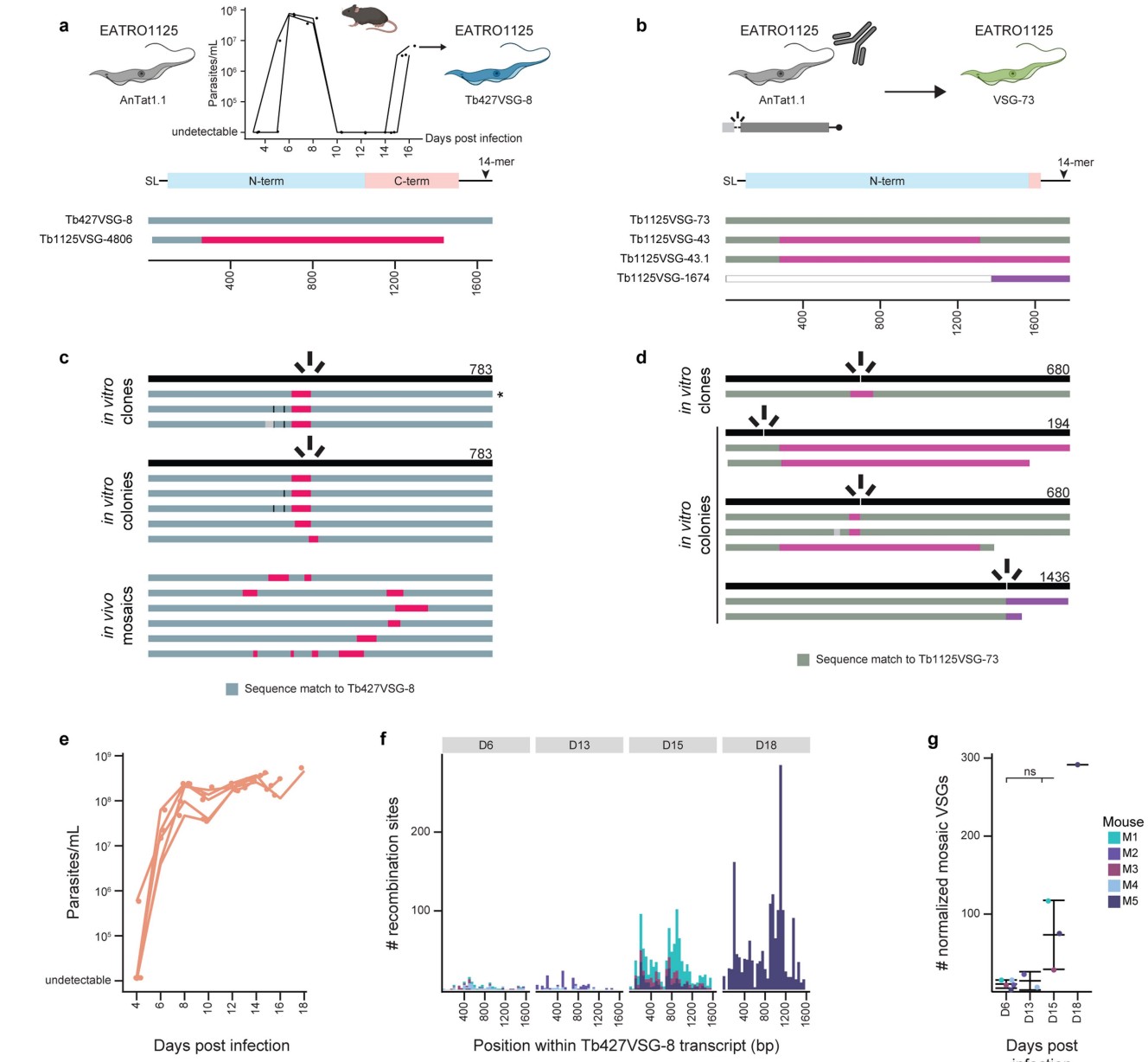

**Extended Data Fig. 5 | Mosaic recombination occurs and can be induced in diverse VSGs.** A) A schematic of isolation of Tb427VSG-8 expressing EATRO1125 parasites. The identified EATRO1125 donor VSG aligned to Tb427VSG-8 shown below. B) A schematic of isolation of Tb1125VSG-73 expressing EATRO1125 parasites. EATRO1125 donor VSGs aligned to Tb1125VSG-73 shown below. The white portion of Tb1125VSG-1674 is significantly different than the VSG-73 sequence. C) Mosaic VSGs observed for Tb427VSG-8 in vitro after a cut with guide 783 and in vivo. Mosaic colonies contained multiple VSGs, including intact Tb427VSG-8 and multiple mosaic species (detailed in Supplemental Excel File 1). Light blue sequences are a match to Tb427VSG-8 sequence. * indicates the population was not perfectly clonal, but the majority of the reads indicate this mosaic VSG. D) Mosaic VSGs observed for Tb1125VSG-73 in vitro following cuts at positions 194, 680, and 1436. Light green sequences are a match to Tb1125VSG-73. E) A time course of the parasitemia for Tb427VSG-8-expressing EATRO1125 parasites in µMT mice. Blood was harvested every two days. F) Histograms of identified mosaic Tb427VSG-8 recombination events using donor Tb1125VSG-4806. Mouse blood was harvested on different days post infection and plotted as such, colored by mouse. G) Quantification of the number of recombinant VSGs identified during the mouse infections normalized by the total number of VSGs in each sample as identified via BLAST divided by the total number of VSGs in a single sample to control for sequencing depth. Statistical significance was determined by a Shapiro-Wilk normality test followed by a two-tailed pairwise Wilcoxon signed-rank test between D6 and D13-D15. Mean ± s.d. Illustrations in **a** and **b** created in BioRender; Smith, J. https://biorender.com/s6686s3 (2026).

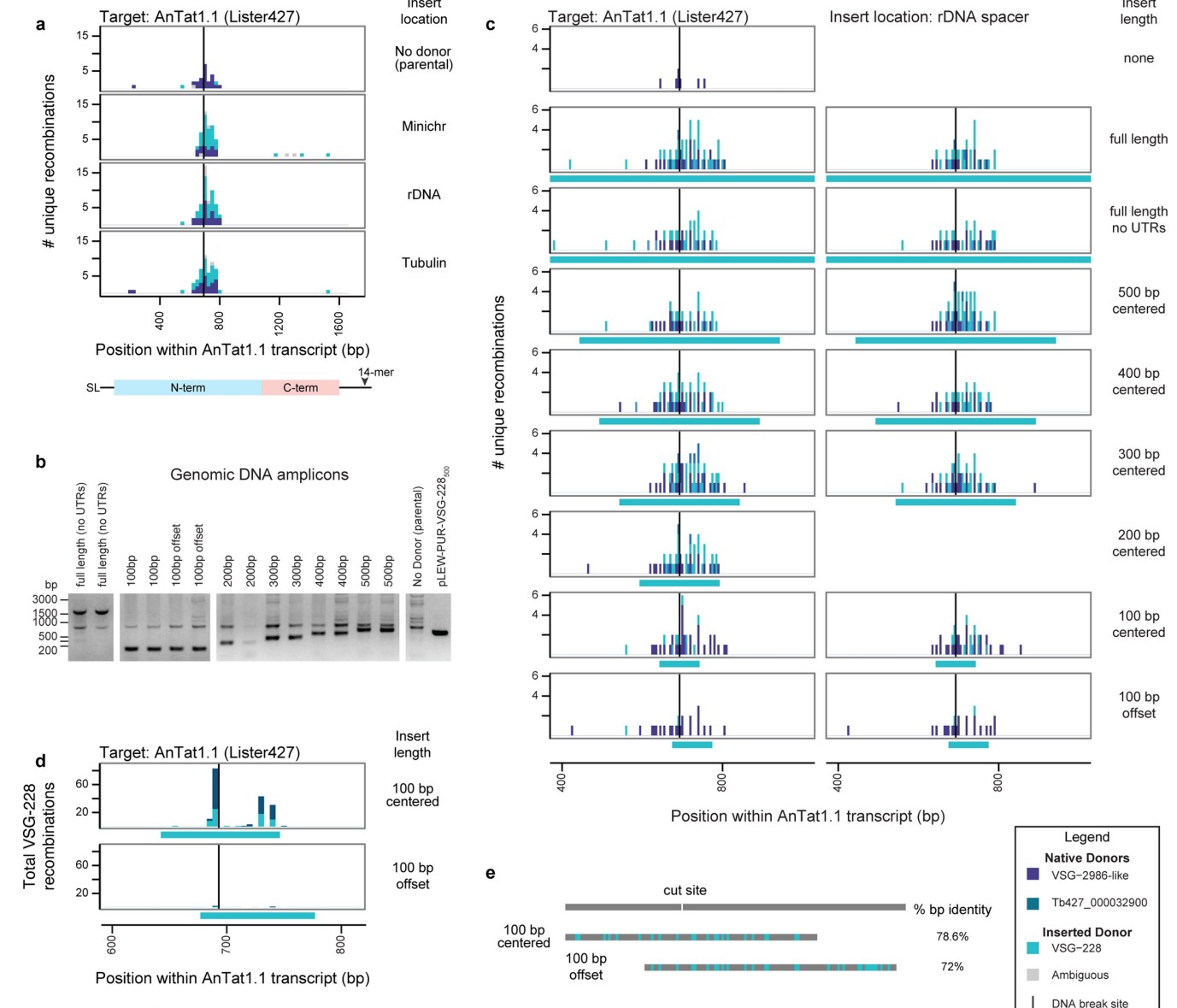

**Extended Data Fig. 6 | Ectopic donor recombination extended data.**
A) Histogram of unique recombination events after a cut at position 694 in AnTat1.1 in AnTat1.1-expressing Lister427 parasites carrying the VSG-228 donor. The midpoint of perfect homology between AnTat1.1 and the donor VSG at the recombination site is plotted. B) PCR of genomic DNA from each clone with an inserted donor VSG-228 to verify insert length. C) Histograms of all the recombination events after a cut at position 694 in AnTat1.1 in Lister427 parasites expressing AnTat1.1. Below each histogram is the aligned VSG-228 donor sequences of various truncated lengths. D) Histogram of all the recombination events after a cut at position 694 in AnTat1.1 in AnTat1.1-expressing Lister427 parasites with 100 bp of VSG-228 donor centered around the cut site or offset by 33 bp. E) Schematic of the 100 bp VSG-228 donor sequences. Blue = unique to VSG-228, gray = shared between AnTat1.1 and VSG-228. Percent identity is shown on the right. SL = 5′ splice leader, 14-mer = 3′ sequence conserved in all VSG transcripts.

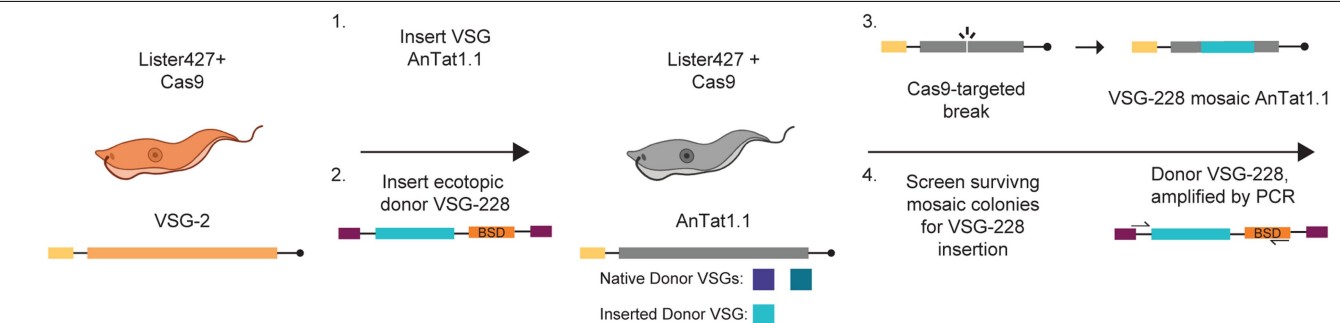

**Extended Data Fig. 7 | Evaluating donor VSG status.** A schematic of the donor template status experiment. Lister427 parasites are modified to express AnTat1.1 with a silent copy of VSG-228. (1 & 2) AnTat1.1 is targeted by Cas9 at position 694 and clones utilizing VSG-228 as a donor were isolated (3 & 4). The donor sequence was amplified from genomic DNA and sequenced (4). Illustrations created in BioRender; Smith, J. https://biorender.com/88kgepj (2026).

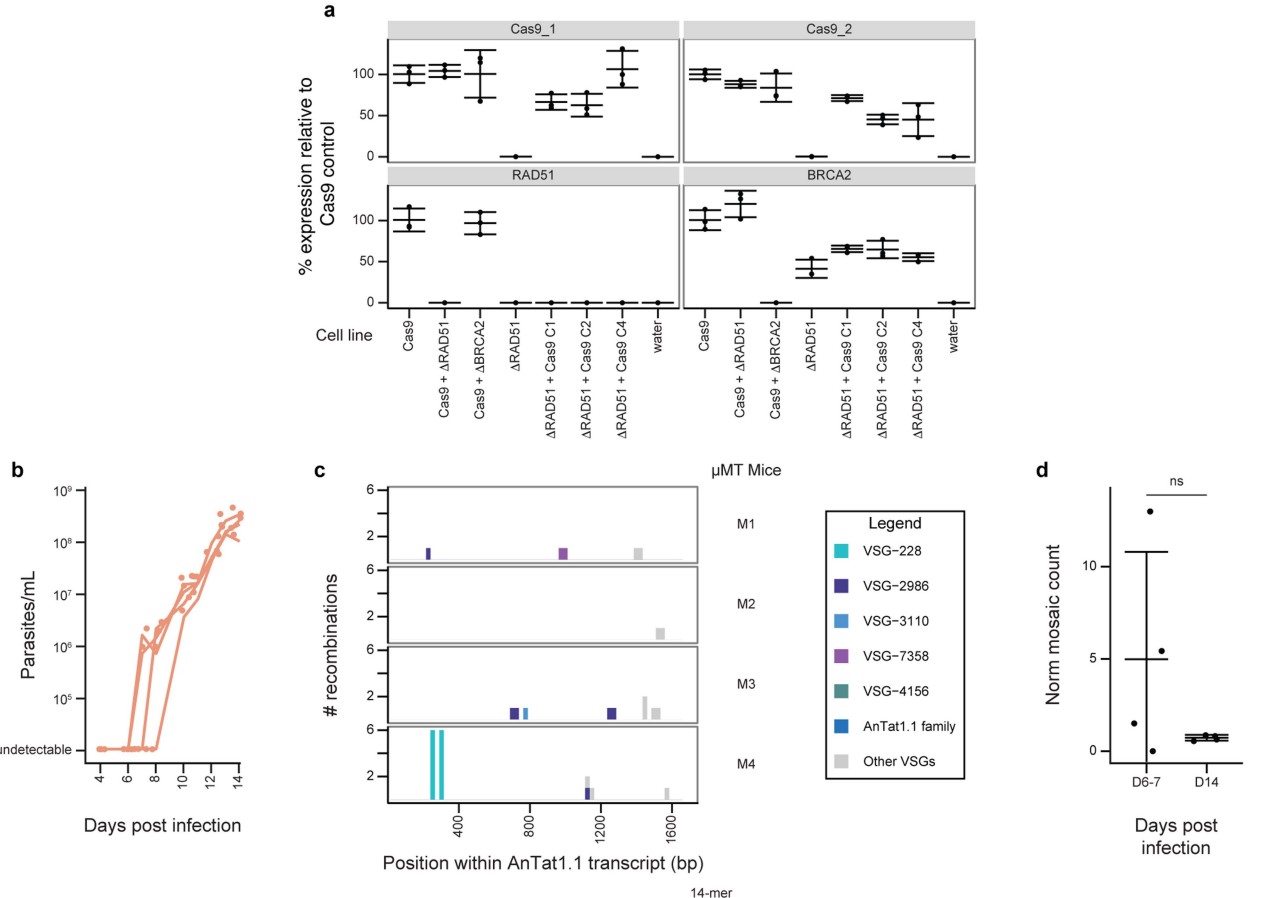

**Extended Data Fig. 8 | RAD51/BRCA2 Extended Data.** A) Quantification of qPCR of RNA from parasite knockout clones confirming Cas9 expression and proper gene knockout. Percent expression was normalized relative to EATRO1125 + Cas9. The clone name indicates the order in which each genetic modification occurred. Mean ± s.d. B) A time course of the parasitemia for ΔRAD51 parasites in μMT mice. Blood was harvested every two days. C) A histogram of all detected recombination events in each mouse. The midpoint of the perfect homology between AnTat1.1 and the donor VSG at the recombination site is plotted. If a mosaic sequence matched >1 potential donor VSG, the average recombination position was plotted. D) Quantification of the number of recombination events detected per mouse. Each mouse was normalized to the number of total aligned, unanchored reads compared to the aligned, unanchored read count for one mouse to control for sequencing depth. Statistical significance was determined by a Shapiro-Wilk normality test followed by a two-tailed by pairwise Wilcoxon signed-rank test. n = 4 biological replicates Mean ± s.d. SL = 5′ splice leader sequence, 14-mer = 3′ sequence conserved in all VSG transcripts.

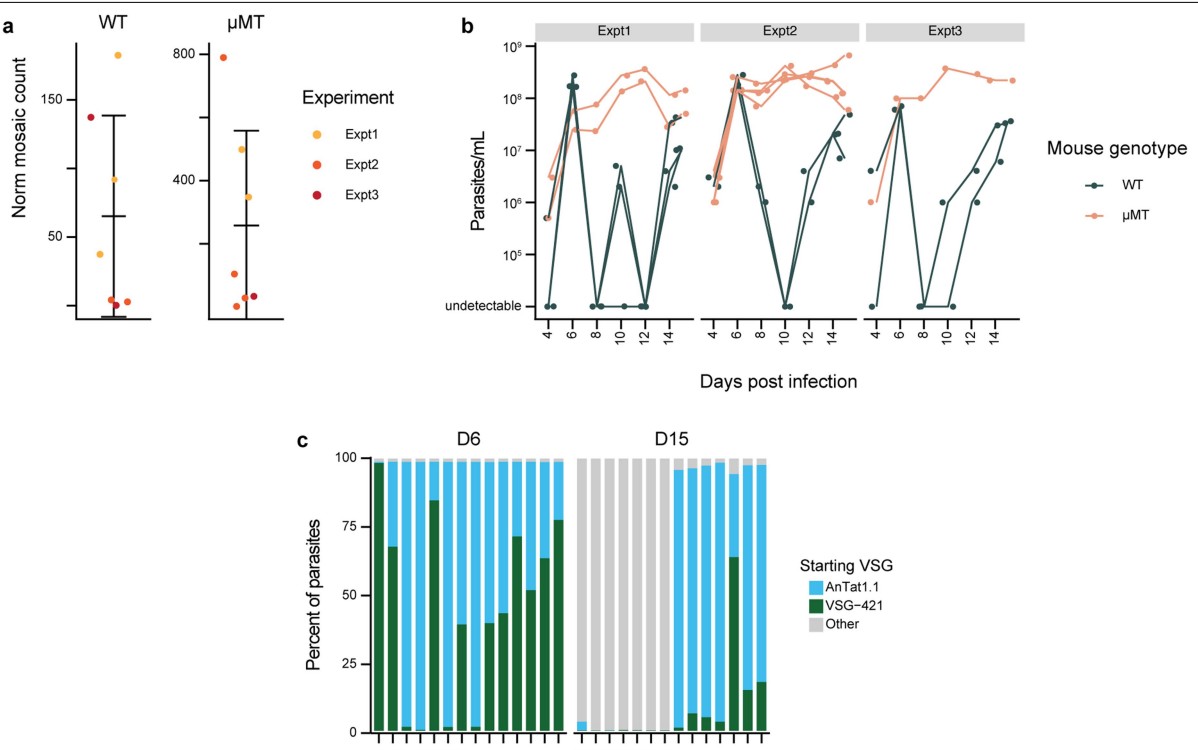

**Extended Data Fig. 9 | Mouse infection extended data.** A) Quantification of the number of recombination events detected per mouse. Each mouse sample was normalized to the number of total consolidated, aligned, and unanchored reads compared to the consolidated, aligned, unanchored read count for one mouse sample to control for sequencing depth. This was performed separately for each genotype. n = 7, each genotype Mean ± s.d. B) A time course of the parasitemia for the mouse infections with wildtype parasites. Blood was harvested every two days. C) Quantification of the percent of parasites expressing the starting VSG at D6 and D15 post-infection as quantified by VSG-seq.

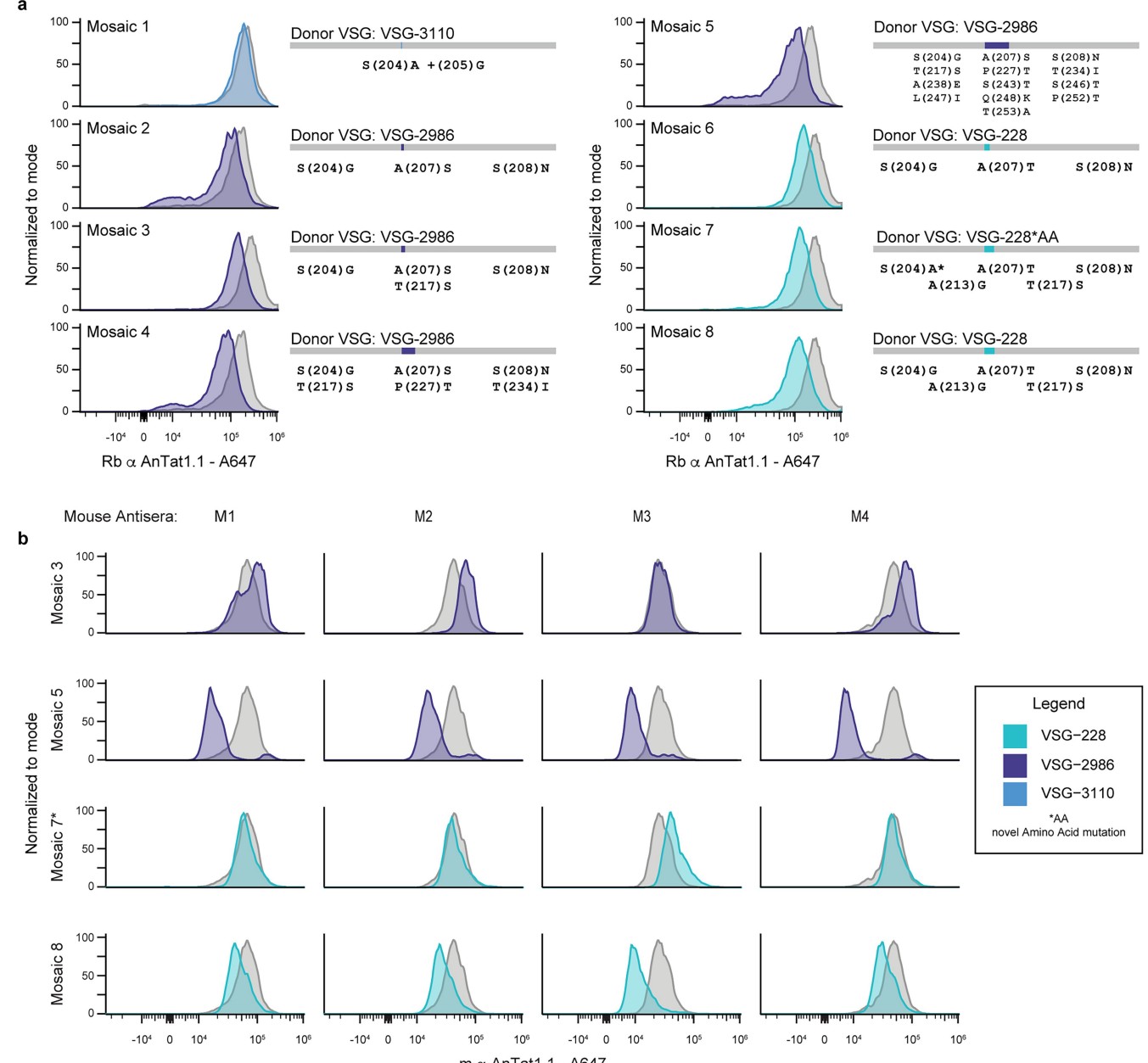

**Extended Data Fig. 10 | Mosaic AnTat1.1 VSGs stained by rabbit and mouse antisera.** A) Histograms showing the binding of anti-AnTat1.1 antibody, as measured by anti-rabbit IgG Alexa Fluor 647 staining intensity, for parental controls (gray) and mosaic clones colored by donor VSG. (n = 4 technical replicates, from 4 independent parental lineages derived from 2 Cas9 clones). Schematics showing the mosaic VSGs are shown on the right with donor VSG and amino acid substitutions specified. Gray sequences are a perfect match to AnTat1.1. B) Histograms showing the binding of mouse antisera generated from AnTat1.1-expressing parasite infections from four mice (M1-M4) as measured by anti-mouse IgG Alexa Fluor 647 staining intensity. AnTat1.1 parasites are in gray and mosaic clones colored by donor VSG. (n = 3 technical replicates, n = 4 biological replicates).

|  |  |
|---|---|

## Reporting Summary

## Statistics

For all statistical analyses, confirm that the following items are present in the figure legend, table legend, main text, or Methods section.

| n/a | Confirmed |  |
|---|---|---|
| ☐ | ☒ | The exact sample size (*n*) for each experimental group/condition, given as a discrete number and unit of measurement |
| ☐ | ☒ | A statement on whether measurements were taken from distinct samples or whether the same sample was measured repeatedly |
| ☐ | ☒ | The statistical test(s) used AND whether they are one- or two-sided *Only common tests should be described solely by name; describe more complex techniques in the Methods section.* |
| ☐ | ☒ | A description of all covariates tested |
| ☐ | ☒ | A description of any assumptions or corrections, such as tests of normality and adjustment for multiple comparisons |
| ☐ | ☒ | A full description of the statistical parameters including central tendency (e.g. means) or other basic estimates (e.g. regression coefficient) AND variation (e.g. standard deviation) or associated estimates of uncertainty (e.g. confidence intervals) |
| ☐ | ☒ | For null hypothesis testing, the test statistic (e.g. *F*, *t*, *r*) with confidence intervals, effect sizes, degrees of freedom and *P* value noted *Give P values as exact values whenever suitable.* |
| ☐ | ☒ | For Bayesian analysis, information on the choice of priors and Markov chain Monte Carlo settings |
| ☐ | ☒ | For hierarchical and complex designs, identification of the appropriate level for tests and full reporting of outcomes |
| ☒ | ☐ | Estimates of effect sizes (e.g. Cohen's *d*, Pearson's *r*), indicating how they were calculated |

*Our web collection on statistics for biologists contains articles on many of the points above.*

## Software and code

Policy information about availability of computer code

| Data collection | For VSG Seq, libraries were sequenced with 100bp single-end reads on a NovaSeq6000. For VSG AMP Seq, reads were sequenced with a MiSeq or NovaSeq6000 (muMT mouse samples) with custom index1 and read2 primers with paired-end reads using the following cycle conditions: "151\|8\|33\|131" . Single parasite clones were sequenced by Plasmidsaurus using oxford nanopore technology and their custom analysis and annotation software. other parasite clones were sequenced with Sanger sequencing using custom primers reported in the manuscript. <br><br> Nanopore sequencing for VSG amplicons from muMT infected mouse blood (originally infected with Tb427VSG-8) was performed with a PromethION using a 10.4.1 flow cell and the SQK-LSK114 kit. |
|---|---|
| Data analysis | The VSG-Seq pipeline (1) is available on github: http://github.com/mugnierlab/VSGSeqPipeline The software versions used were: Trinity(2.8.5), Biopython(1.72), Blast(2.9), Bedtools(2.29.2), cd-hit(4.8.1), trim-galore(0.6.4), bowtie(v1.2.3), and samtools(1.9). <br><br> VSG AMP Seq pipeline was created for custom analysis of the VSG AMP Seq reads. The software is available on http://github.com/mugnierlab/Smith2026/tree/main/VSG-AMP-Seq. A module of VSG-AMP-seq was also deposited onto CodeOcean: 10.24433/CO.3005244.v1 and 10.24433/CO.3283350.v1. The software versions used were: Python(3.8.19), biopython(1.78), cutadapt(3.5), trim-galore(v0.6.4), pandas(1.2.1), python-levenshtein(0.25.1), progress-bar(2.5), regex(2.5.82), bowtie(1.3.1), cd-hit(v4.8.1). Modules are manually run via: index_addition, primer_sort, trim, demultiplex, consol_reads, vsg_align, define_read_consensus, identify_mosaics. The output of this can be run through custom R functions included in mosaic_graphing_functions. Analysis was run with R(4.0.2). Quantification of reads uses quant_R1s_bowtie. To run this pipeline custom VSG-specific files and experiment-specific are required including: global_target.py, a _primer.txt file, a _primers.fasta file, a fasta file containing the target sequence, and a custom barcodes file. |

VSGnome analysis can be run with the pipeline genome_VSG_analysis. It requires blast(2.10.0). VSG structural prediction was performed by isolating N-termini with the following available scripts: github.com/mugnierlab/find_VSG_Ndomains. It requires software SignalP(6.0), HMMscan(3.1b2), and ColabFold(1.5.5). Visualization used UCSF ChimeraX (1.7.1).

Scripts to identify the in vivo VSG-8 sequences can be found at github.com/mugnierlab/smith2025/tree/main/altVSG8_nanopore_analysis. The software versions used were: Dorado (7.2.13), samtools (1.22.1), cutadapt (4.9), sed (4.5), hisat2 (2.1.0), and BLAST(2.16.0+).

Scripts to reconstruct mosaic VSGs from mixed nanopore populations can be found at github.com/mugnierlab/smith2026/tree/main/nanopore_colony_consensus_builder. The software versions used were: sed(4.5), cd-hit(4.8.1), minimap2(2.30-r1287), and samtools(1.22.1).

FlowJo(10.8.1) was used to perform gating and analyze flow cytometry data.

Western blots were analyzed with FIJI(2.14.0/1.54f). Code used to generate all figures is found at github.com/mugnierlab/Smith2026/tree/main/Figures.

For manuscripts utilizing custom algorithms or software that are central to the research but not yet described in published literature, software must be made available to editors and reviewers. We strongly encourage code deposition in a community repository (e.g. GitHub). See the Nature Portfolio guidelines for submitting code & software for further information.

## Data

Policy information about availability of data

All manuscripts must include a data availability statement. This statement should provide the following information, where applicable:
- Accession codes, unique identifiers, or web links for publicly available datasets
- A description of any restrictions on data availability
- For clinical datasets or third party data, please ensure that the statement adheres to our policy

Data for generating figures, i.e. the output of VSG-AMP-seq, are available on github: https://github.com/mugnierlab/Smith2025/tree/main/Figures. The raw sequencing reads for all data are available in the National Center for Biotechnology Information (NCBI) Sequence Read Archive under accession number PRJNA1140873.

All mosaic VSGs isolated from individual clones, and donor VSG amplicon sequences with cooresponding mosaic VSG sequences are on github.com/mugnierlab/Smith2026.

VSG-seq from in vivo mouse infections (blood & tissues) is found in Beaver et al. and at github.com/mugnierlab/Beaver2022.

VSG sequences for analysis are found at tryps.rockefeller.edu/Sequences.html. Additional VSG sequences were identified in TriTrypDB release 66, Lister2018 genome.

VSG N-terminal  domain HMM profile can be found at github.com/mugnierlab/find_VSG_Ndomains.

# Research involving human participants, their data, or biological material

Policy information about studies with human participants or human data. See also policy information about sex, gender (identity/presentation), and sexual orientation and race, ethnicity and racism.

| | |
|---|---|
| Reporting on sex and gender | n/a |
| Reporting on race, ethnicity, or other socially relevant groupings | n/a |
| Population characteristics | n/a |
| Recruitment | n/a |
| Ethics oversight | n/a |

Note that full information on the approval of the study protocol must also be provided in the manuscript.

# Field-specific reporting

Please select the one below that is the best fit for your research. If you are not sure, read the appropriate sections before making your selection.

☒ Life sciences          ☐ Behavioural & social sciences          ☐ Ecological, evolutionary & environmental sciences

For a reference copy of the document with all sections, see nature.com/documents/nr-reporting-summary-flat.pdf

# Life sciences study design

All studies must disclose on these points even when the disclosure is negative.

| Sample size | Based on our previous work, a sample size of 4-5 mice is typically sufficient for analyses of antigenic variation, so we sought to achieve this sample size for all experiments.<br> (Mugnier MR, Cross GA, Papavasiliou FN. The in vivo dynamics of antigenic variation in Trypanosoma brucei. Science. 2015 Mar 27;347(6229):1470-3. doi: 10.1126/science.aaa4502. PMID: 25814582; PMCID: PMC4514441<br>Beaver, A.K., Keneskhanova, Z., Cosentino, R.O. et al. Tissue spaces are reservoirs of antigenic diversity for Trypanosoma brucei. Nature 636, 430–437 (2024). https://doi.org/10.1038/s41586-024-08151-z)<br><br>muMT mice, as they are immunocompromised, sometimes died prematurely before they reached the second corresponding peak of parasitemia in the WT mice. We infected enough mice (15 of each mouse genotype) such that sufficient muMT mouse survival across all 3 infections could ensure sufficient sample size. Experiments resulted in n=7 muMT and n=7 WT mice, across 3 separate infections. For subsequent experiments with muMT mice, we infected 5 mice with parasites. Here when we infected mice with RAD51-knockout parasites, we ended up with n=4, but are appropriately tentative with our conclusions from the results. VSG-8 expressing infections in muMT mice were n=5 with some mice dying before D15, like in infections with AnTat1.1. We were able to collect samples from all of these mice, extending our time course over multiple time points. Sample size at each time point limits the statistical significance of the number of mosaic VSGs observed, however these infections were sufficient to identify numerous mosaic VSGs generated in vivo. |
|---|---|
| Data exclusions | We excluded additional WT mice which survived - such that the number of WT mice and the number of muMT mice matched. This was chosen randomly among the samples - across the 3 experiments. |
| Replication | These experiments represent 7 biological replicates of each genotype across 3 independent experiments. Each infection was started at a distinct time with a distinct vial of parasites. muMT and WT mice were infected within an experiment at the same time. RAD51-knockout and VSG-8 mouse experiments were performed once, but with 4-5 biological replicates, from one vial of parasites. No additional attempts were made to perform these experiments beyond what is described here. |
| Randomization | Mice were housed together based on mouse genotype. Mice were randomly numbered 1-5 and treated in that order throughout the experiment. muMT mice were treated and handled first to prevent secondary infections. Sample processing occurred in batches based on experiment, but included both genotypes. |
| Blinding | Blinding of the samples was not necessary for the investigators since each sample was processed through a pipeline which treated them all equivalently. In addition, all wildtype parasite samples were sequenced with both VSG-Seq and VSG-AMP-Seq where results matched. |

# Reporting for specific materials, systems and methods

We require information from authors about some types of materials, experimental systems and methods used in many studies. Here, indicate whether each material, system or method listed is relevant to your study. If you are not sure if a list item applies to your research, read the appropriate section before selecting a response.

## Materials & experimental systems

| n/a | Involved in the study |
|---|---|
| ☐ | ☒ Antibodies |
| ☐ | ☒ Eukaryotic cell lines |
| ☒ | ☐ Palaeontology and archaeology |
| ☐ | ☒ Animals and other organisms |
| ☒ | ☐ Clinical data |
| ☒ | ☐ Dual use research of concern |
| ☒ | ☐ Plants |

## Methods

| n/a | Involved in the study |
|---|---|
| ☒ | ☐ ChIP-seq |
| ☐ | ☒ Flow cytometry |
| ☒ | ☐ MRI-based neuroimaging |

## Antibodies

| Antibodies used | We used a custom AnTat1.1 polyclonal CRD-depleted (cross-reactive domain) antibody provided by Jay Bangs. Other antibodies used included: mouse anti FLAG (M2) monoclonal antibody (Millipore Sigma, F3165-1MG), mouse anti EF1a (CBP-KK1 clone) (Millipore 903 Sigma, 05-235), and rabbit anti g-H2A was a kind gift from Galadriel Hovel-Miner based upon Glover and Horn. We also generated mouse polyclonal antisera against AnTat1.1 expressing Lister427 parasites. We had 4 biological replicates, M1-M4. Secondary antibodies for western blot were: goat anti mouse (Cell Signaling, 7076S) or goat anti rabbit-HRP-conjugated secondary (Cell Signaling, 7074S). Secondary antibodies for flow cytometry were: Alexa Fluor 647-conjugated goat anti rabbit IgG (H+L), 911 F(ab')2 Fragment (Cell Signaling, 4414S) and goat anti mouse IgG (H+L), F(ab')2 Fragment (Cell Signaling, 4410S). |
|---|---|
| Validation | The AnTat1.1 polyclonal antibody has been validated in Beaver et al, 2024, but also used to stain AnTat1.1 expressing parasites and Lister427 parasites which express VSG-2. Positive Signal was only observed for the parasites expressing AnTat1.1. Live/dead propidium iodide staining was validated using heat killed parasites. g-H2A was detailed in Glover and Horn Ref: Glover, L. & Horn, D. Trypanosomal histone gH2A and the DNA damage response. Mol Biochem Parasitol. (2012)183(1):78-83. This was validated by a peptide competition assay. AnTat1.1 expressing Lister427 antisera from the mice was stained against intact AnTat1.1 expressing parasites and VSG-228 which is antigenically distinct. These are in figure 5. |

No statements were made for mouse anti FLAG M2 on the manufacturer's website. Western blot signal is only observed in trypanosome lysate when recombinant FLAG-tag protein is introduced.
mouse anti EF1a is validated for use in IP and WB per the manufacturer. This clone was partially validated here:
Kaur KJ, Ruben L. Protein translation elongation factor-1 alpha from Trypanosoma brucei binds calmodulin. J Biol Chem. 1994 Sep 16;269(37):23045-50. PMID: 8083206.

The following publications have used CBP-KK1 as a loading control for Trypanosoma brucei western blots:
Keneskhanova, Z., McWilliam, K.R., Cosentino, R.O. et al. Genomic determinants of antigen expression hierarchy in African trypanosomes. Nature 642, 182–190 (2025). https://doi.org/10.1038/s41586-025-08720-w
Escrivani DO, Scheidt V, Tinti M, Faria J, Horn D. Competition among variants is predictable and contributes to the antigenic variation dynamics of African trypanosomes. PLoS Pathog. 2023 Jul 17;19(7):e1011530. doi: 10.1371/journal.ppat.1011530. PMID: 37459347; PMCID: PMC10374056.

# Eukaryotic cell lines

Policy information about cell lines and Sex and Gender in Research

| Cell line source(s) | EATRO 1125 AnTat1.1 90-13 T. brucei (gifted from Keith Mathews, Ref: Engstler, M. & Boshart, M. Cold shock and regulation of surface protein trafficking convey sensitization to inducers of stage differentiation in Trypanosoma brucei . Genes Dev. 18, 2798(2004)) <br> Monomorphic Single Marker Lister427 VSG221 TetR T7RNAP bloodstream form (NR42011; Lot: 61775530) Wirtz, E. Leal, S., Ochatt, C. & Cross, G.A.M. A tightly regulated inducible expression system for conditional gene knock-outs and dominant negative genetics in Trypanosoma brucei. Mol. Biochem. Parasitol. 99, 89-101(1999) <br><br> Monomorphic Single Marker 427 1339 Cas9 TetR T7RNAP (bloodstream form) (NR-56793; LOT: 70056027) <br><br> EATRO1125 AnTat1.1 J1339 pleiomorphic parasites were a gift from Keith Matthews. |
| --- | --- |
| Authentication | Cell lines were authenticated via sequencing of the VSG expressed by the parasite and PCR of drug resistance markers and corresponding drug resistance. |
| Mycoplasma contamination | Cells lines were not confirmed for mycoplasma contamination |
| Commonly misidentified lines (See ICLAC register) | No commonly misidentified lines were used. |

# Animals and other research organisms

Policy information about studies involving animals; ARRIVE guidelines recommended for reporting animal research, and Sex and Gender in Research

| Laboratory animals | C57Bl/6J (WT, strain # 000664 Jackson Laboratory) or B6.129S2-Ighmtm1Cgn/J ( μMT-, strain #002288 Jackson Laboratory) between 8-12 weeks were used for the study. Mouse were injected intravenously in the tail vein with 5 parasites or IP with 100 parasites for antibody generation. |
| --- | --- |
| Wild animals | No wild animals were used in this study. |
| Reporting on sex | Only female mice were used for this study. Previous experiments in the field suggest sex does not have a major influence on antigenic variation. Cost also limited our ability to perform this experiment in multiple sexes. |
| Field-collected samples | No field collected samples were used in this study. |
| Ethics oversight | All animal studies were approved by the Johns Hopkins Animal Care and Use Committee (protocol # MO22H163). |

Note that full information on the approval of the study protocol must also be provided in the manuscript.

# Plants

| Seed stocks | *Report on the source of all seed stocks or other plant material used. If applicable, state the seed stock centre and catalogue number. If plant specimens were collected from the field, describe the collection location, date and sampling procedures.* |
| --- | --- |
| Novel plant genotypes | *Describe the methods by which all novel plant genotypes were produced. This includes those generated by transgenic approaches, gene editing, chemical/radiation-based mutagenesis and hybridization. For transgenic lines, describe the transformation method, the number of independent lines analyzed and the generation upon which experiments were performed. For gene-edited lines, describe the editor used, the endogenous sequence targeted for editing, the targeting guide RNA sequence (if applicable) and how the editor was applied.* |
| Authentication | *Describe any authentication procedures for each seed stock used or novel genotype generated. Describe any experiments used to assess the effect of a mutation and, where applicable, how potential secondary effects (e.g. second site T-DNA insertions, mosiacism, off-target gene editing) were examined.* |

# Flow Cytometry

## Plots

Confirm that:

☒ The axis labels state the marker and fluorochrome used (e.g. CD4-FITC).

☒ The axis scales are clearly visible. Include numbers along axes only for bottom left plot of group (a 'group' is an analysis of identical markers).

☒ All plots are contour plots with outliers or pseudocolor plots.

☒ A numerical value for number of cells or percentage (with statistics) is provided.

## Methodology

| | |
|---|---|
| Sample preparation | In 96 well plates, 200,000 parasites were stained with 1:20,000 rabbit anti AnTat1.1 primary antibody (Jay Bangs)  or 1:1000 mouse anti AnTat1.1 serum for ten minutes at 4C while shaking in PBS + 10mg/mL glucose. Parasites were washed once with 100uL PBS + glucose. Then, parasites were stained with Alexa Fluor 647-conjugated goat anti rabbit IgG (H+L), F(ab')2 Fragment (Cell Signaling, 4414S) or goat anti mouse IgG (H+L), F(ab')2 Fragment (Cell Signaling, 4410S) at 1:1000 at 4C while shaking in PBS + glucose. After washing again with 100uL PBS + glucose, parasites were resuspended in PBS + glucose + 1:20 Propidium Iodide (BD Biosciences, 556463) and analyzed on a Attune Nxt Flow cytometer (Invitrogen). Data analysis was performed using FlowJo v10. |
| Instrument | Attune Nxt Flow Cytometer (Invitrogen) |
| Software | FlowJo (10.6.1) |
| Cell population abundance | Each parasite was obtained from a single cell clone, in a limited dilution 96-well plate such that less than 30 clones were present. The VSG expressed by these parasites was sequenced, and purity of individual sequencing reads was assessed to ensure just one VSG was expressed by the clone and it had not arisen from two clones, or a subpopulation had undergone a switch. Parasites were counted before they were plated and a similar number of parasites was run through the instrument to assess global staining of the parasites. |
| Gating strategy | For flow cytometry experiments, we first gated on T. brucei cells using FSC-A vs SSC-A. Next, we gated on single cells using FSC-A vs FSC-H. Finally, we gated on alive cells using the PI negative staining using the compensated Alexa Fluor 647-A vs compensated PI-A. Alive cells were gated and analyzed in histograms in Figure 6. |

☒ Tick this box to confirm that a figure exemplifying the gating strategy is provided in the Supplementary Information.

