## [Peer Review File · Nature]

DNA damage drives antigen diversification in *Trypanosoma brucei*

Corresponding Author: Dr Monica Mugnier

Version 0:

Reviewer comments:

Referee #1

(Remarks to the Author)

The manuscript from Smith and colleagues describes the development of a new method to systematically investigate the role of DNA double strand break repair in the generation of new surface antigen proteins in African trypanosomes. African trypanosomes include numerous species of infectious protozoans that cause disease in both humans and livestock as well as wild African animals. They also represent a fascinating system for the study of nucleic acid biology that is evolutionarily very distant from typical model organisms. Here the authors study the process of recombination within the large gene family that encodes the variant surface glycoprotein (VSG) that coats the surface of the parasite. These recombination events enable the parasites to avoid antibody recognition and maintain chronic infections that are typically fatal if not treated. VSG recombination has been studied extensively over the last several decades although the precise details of the process are not yet fully understood. Here the authors apply CRISPR-Cas9 technology to induce double strand breaks with an expressed vsg copy and examine the repair products en masse. This enabled them to study the products of repair in much greater detail than was previously possible and has provided them with a powerful system to study this process.

Overall, the manuscript provides a significant step forward for those in the field interested in the process of VSG diversification and the study of DNA double strand break repair. They demonstrate that an induced break within the vsg coding region results in repair through homologous recombination as opposed to breaks elsewhere within the expression site that induce switching. They further show that repair is based on sequence homology and that the repair products resemble those observed during an infection. Their results also suggest a RAD51-independent recombination mechanism. Many of these conclusions had been suggested previously, however this is the first time these models have been systematically tested through the recovery of large-scale observations using a high throughput system. The ability to trigger this process "on demand" provides a very powerful tool for future studies.

Specific question for the authors to consider:

1. The initial set of experiments employed the parasite line EATRO1125, which express the VSG AnTat1.1. For readers unfamiliar with this line, it would be good to know if these parasites are able to switch to expression of an alternative expression site or if they have been "fixed" to expression only this locus (as is occasionally done through the insertion of a selectable marker within one of the ESAGs). This is relevant given that previous studies have described expression site switching in response to double strand breaks.
2. I am curious how efficient the system is in creating DNA breaks. On lines 207-214 the authors describe obtaining 29 mosaic-expressing clones. How many clones were isolated and what percentage were not expressing a mosaic vsg? In those not expressing a mosaic vsg (if there were any), what were they expressing?
3. Similar to point 2 above, on lines 236-245, the authors describe the results of inducing breaks in VSG-2. No mosaic vsgs were observed and the clones were no longer expressing VSG-2. Had these parasites undergone expression site switching or had there been a recombination event that replaced the entire VSG-2 coding region?

4. The authors frequently express surprise that the parasites were able repair a double strand break using homologous sequences from elsewhere in the genome, implying a “sophisticated homology search mechanism” and that this was “particularly unexpected”. I gather that the authors expected that the typical organization of the vsg repertoire would play a role in homology search. However, this type of homology search is quite typical for homologous recombination in virtually all eukaryotes, thus many readers will not find this in any way unexpected. If this is a truly surprising find, some additional elaboration would be helpful.

5. The authors report that they observed more mosaic VSGs in parasites obtained from extravascular spaces and from this they infer that mosaics might form more frequently in this niche. Is this stated correctly? It seems equally or more likely that they form at the same rate in the extravascular spaces but that they are cleared less efficiently.

6. In the Discussion section, the authors suggest that possibility that they are observing RAD51-independent recombination during the creation of mosaic VSGs. If this process is in fact not typical homologous recombination, is it possible to check to make sure that the donor sequence remains fully intact, as would be expected for a typical gene conversion event?

(Remarks on code availability)

Referee #2

(Remarks to the Author)

Trypanosoma brucei can maintain an infection of large mammalian hosts for years despite being constantly exposed to the host immune system. The persistence of an infection is facilitated by the evolution of a population survival strategy based on antigenic variation of the VSG protein. The adaptive immune system is able to kill trypanosomes once there is sufficient VSG antibody titre but a small number of trypanosomes have switched VSG and, if antigenically novel, escape killing until recognised by antibodies in turn. At the level of the genome, VSG switching can occur through an epigenetic mechanism in which one VSG promoter is silenced and another activated, but this appears restricted to the first few weeks of an infection. The majority of switches occur through gene conversion using a reservoir of VSG sequences that comprise >25% of the genome. The gene conversion events are not simple cassette switches analogous to the budding yeast mating type switch but are complicated usually involving several donor sequences and can generate novel VSGs. This is important as there may not be sufficient VSG coding sequence in the genome to sustain multi-year infections. The low frequency of these gene conversion events mean that the molecular processes involved have been extrapolated from before and after pictures.

In this manuscript, the authors have developed a method to systematically capture individual gene conversion events after introducing a double stranded break in the active VSG gene. The subsequent gene conversion events were found to be similar to those occurring over a time course in long term infections in mice. The experiments have allowed the authors to determine the length of the gene conversion, the length of homology required, and a positional bias.

The sequencing strategy used is novel and effective. The originality and significance of the work is that it provides a holistic view of the gene conversion events occurring with and without immune selection to single base resolution, this is the first time such events have been described in such detail.

The conclusions are generally robust. There is one point that should be discussed and that is whether antibodies just recognise the top of the VSG. It may be worth including thoughts on PMID: 8821936 in the discussion.

Overall this is a complete set of experiments that provide important new information on the mechanisms of gene conversion in VSG antigenic variation. It is exceptionally well written and is a model of clarity.

Some minor points

Line 103

‘two highly repetitive and damage prone stretches of DNA’

The references (2 and 18) probably show they are repetitive but do they show they are damage prone? Is it any more damage prone than an rRNA locus?

Line 209

A brief explanation of why 7 days induction was used?

Line 496

A comment of whether the NHEJ pathway is absent in the majority of trypanosome species? Especially those that do not use VSG based antigenic variation?

(Remarks on code availability)

Referee #3

(Remarks to the Author)

Overview:

Several pathogens, including African trypanosomes undergo antigenic variation to evade the host immune response and rely on genomic diversification to extend their antigenic repertoire.

Many studies have demonstrated that novel VSGs are generated during infections, particularly in late stages of infection. These are thought to emerge through segmental gene conversion ('mosaic' VSGs) or de novo point mutations.

Indeed, the contribution of VSG pseudogenes to the formation of novel 'mosaic' VSGs has long been known (PMID: 15463711; PMID: 1380125; PMID: 2574459). In my view, the scale and complexity of the segmental gene-conversion events associated with the generation of functional 'mosaic' VSGs are clear (PMID: 17652423; PMID 23853603; PMID: 25814582; etc). However, what remains elusive is how this is catalysed.

Studying such mechanisms has been incredibly challenging due to the lack of an in vitro system to do so. Therefore, I congratulate the authors (Smith et al.) on the generation of an efficient and reproducible in vitro system to study this process in a controlled manner, which will no doubt enable hypothesis-driven exploration of mechanisms underpinning antigen diversification in *T. brucei* (and potentially other pathogens).

However, I am not convinced that as it stands this work represents a groundbreaking discovery, since the DNA repair mechanisms involved in mosaic antigen formation remain elusive. I find that most of the observations are consistent with previous work, which at times seems insufficiently credited, and do not add significant mechanistic insight.

Please find specific comments below.

Major Points:

1- The authors state that the frequency of mosaic formation decreased when moving further away from the centre of the AnTat1.1 coding sequence and excluded that this could be due to the guide cutting efficiency based on the results depicted in Fig. S2H. There is a high variability for some of the gRNAs, therefore the data is not very compelling and raises the question whether some of the differences can indeed be due to different guide cutting efficiency. Also, can anti gamma-H2A expression be used as a reliable measurement for guide cutting efficiency? What is the evidence for this?

2- Following on point 1, all the experiments were performed with AnTat1.1, for which mosaics were reported as far back as 1983 (Pays et al, Cell, 1983, PMID 6616615). Have you tried to look at other VSGs that are members of VSG families? Do you obtain the same pattern of recombination events in vitro and in vivo?

3- As mentioned above in the overview paragraph, the scale and complexity of the segmental gene-conversion events that underpin the generation of functional 'mosaic' VSGs are clear (PMID: 17652423; PMID 23853603; PMID: 25814582; etc). However, what remains elusive is the molecular mechanism. The authors have developed an efficient and reproducible in vitro system to study this process in a controlled manner, but their observations do not extend much beyond what was already known apart from the very interesting observation that donor VSGs can be used as templates irrespectively of their genomic localisation. In my opinion, a real breakthrough would have been determining the DNA repair mechanisms underpinning mosaic formation, which the authors do not demonstrate.

Smith et al speculate that the recombination events they describe are likely RAD51-independent, but have they tested this in RAD51 KO cells? An interesting possibility is microhomology-mediated end-joining (MMEJ), which has been previously suggested to contribute (Glover et al, NAR, 2011, PMID: 20965968), and the authors also mention this in the discussion. Why not test this?

4- The authors make the bold statement that 'small changes in the VSG sequence provide substantial antibody evasion' (line 351) and 'substantial immune evasion' (lines 457-458). However, what they do show is reduced antibody binding, not immune evasion, which by their own admission in the discussion might not be sufficient. I quote "However, a 50% drop in binding may not be sufficient to fully evade preexisting anti-VSG antibody." Can you show that these differences in antibody binding lead to immune evasion? Also, how are these observations novel or different from what has been previously reported by others, among those Hall et al, PLoS Pathogens, 2013, PMID 23853603 (see Figure 6 specifically)?

Citation and/or discussion of other publications:

1- In the introduction you mention "While mosaic VSGs have been observed in the literature, they are often found under extreme experimental conditions..." – can you define 'extreme experimental conditions'?

2- Also in the introduction, the authors mention that 'DNA breaks are frequently observed within the 70bp repeats at the active and near telomeres within silent BESS', but since they induce breaks in the VSG ORF, this paper is relevant: Girasol et al (PNAS, 2023, PMID: 3798847) show naturally occurring breaks at the VSG-2 ORF in BES1 L427 (repaired by RAD51) using BLISS.

3- When the authors conduct experiments in L427 (Fig. S3), how does the observed switching pattern align with the original experiments conducted by Escrivani et al (PMID 37459347)? This is a paper the authors do not cite and that uses the same approach to trigger VSG switching in L427.

4- I think there are several papers that although were cited in the introduction, should have discussed, a striking one being: Hall et al, PLoS Pathogens, 2013, PMID 23853603. Although the in vitro method and the combination of both in vitro and in

vivo experiments are overall commendable, it is not entirely clear what are the novel findings in this paper in relation to previous studies.

Other Points:

1- In the M&M, the authors state that "Mosaic reads which did not overlap to generate a consensus and did not contain an identifiable recombination site in read1 were excluded from analysis." What is the % of reads that were discarded in this process?

2- In Fig. 2A, is the data for a single clone per gRNA or a combination of 2 clones? If it is a single clone, how reproducible are the donor VSGs between clones? Could you show the 2 clones per gRNA separately (in Supp for instance)?

3- Are you sure the graph in Fig. 2G is not duplicated in Fig. S3A? Or is it just very consistent?

(Remarks on code availability)

Version 1:

Reviewer comments:

Referee #1

(Remarks to the Author)

The revised manuscript from Smith and colleagues describes the development of a new method to systematically investigate the role of DNA double strand break repair in the generation of new surface antigen proteins in African trypanosomes. A mechanistic, molecular understanding of this process has been the focus of numerous laboratories of the last few decades, and this paper provides significant insights into the process.

The original version of the manuscript was well written and the experiments insightful. The authors have improved the manuscript by clarifying their interpretations of several key experiments and by providing additional data, for example studies with Rad51/BRCA2 knockout lines. The statistical analyses appear appropriate and correctly performed. I have no additional requests or suggestions for the authors.

(Remarks on code availability)

The authors provide both the code and a readme file with directions for installation and use of the code.

Referee #3

(Remarks to the Author)

I sincerely thank the authors for the substantial amount of additional work they performed to address the points I raised as well as their thorough response. The revisions—particularly the new mechanistic data—significantly elevate the impact of this work.

Key strengths of the revised manuscript:

- The discovery that mosaic formation is RAD51- and BRCA2-dependent is highly significant. By demonstrating that this process relies on Homologous Recombination (HR) rather than MMEJ, the authors have provided the first definitive identification of the cellular machinery involved.
- Establishing that 50–100 bp of imperfect homology is sufficient for recombination provides essential experimental data on the sequence requirements for diversification.
- The evidence supporting "progressive mosaicism" and the observation of these events in tissue spaces clarify how complex mosaics likely accumulate over time.
- The updated serological assays using mouse antisera—showing up to a 75% reduction in binding—more convincingly supporting that small-scale segmental changes provide meaningful immune evasion.

The authors have successfully addressed my concerns regarding novelty and mechanism. This work provides both a critical new toolkit and fundamental insights into *T. brucei* antigenic variation, representing a major advancement in the field. I unreservedly recommend it for publication in Nature.

(Remarks on code availability)

Response to Reviewers for **DNA Damage drives antigen diversification through mosaic Variant Surface Glycoprotein (VSG) formation in *Trypanosoma brucei***

Referee expertise:

Referee #1: VSGs, Trypanosome mol. bio.

Referee #2: VSGs, Trypanosome mol. bio.

Referee #3: VSGs, Trypanosome mol. bio.

Referees' comments:

Referee #1 (Remarks to the Author):

The manuscript from Smith and colleagues describes the development of a new method to systematically investigate the role of DNA double strand break repair in the generation of new surface antigen proteins in African trypanosomes. African trypanosomes include numerous species of infectious protozoans that cause disease in both humans and livestock as well as wild African animals. They also represent a fascinating system for the study of nucleic acid biology that is evolutionarily very distant from typical model organisms. Here the authors study the process of recombination within the large gene family that encodes the variant surface glycoprotein (VSG) that coats the surface of the parasite. These recombination events enable the parasites to avoid antibody recognition and maintain chronic infections that are typically fatal if not treated. VSG recombination has been studied extensively over the last several decades although the precise details of the process are not yet fully understood. Here the authors apply CRISPR-Cas9 technology to induce double strand breaks with an expressed vsg copy and examine the repair products en masse. This enabled them to study the products of repair in much greater detail than was previously possible and has provided them with a powerful system to study this process.

Overall, the manuscript provides a significant step forward for those in the field interested in the process of VSG diversification and the study of DNA double strand break repair. They demonstrate that an induced break within the vsg coding region results in repair through homologous recombination as opposed to breaks elsewhere within the expression site that induce switching. They further show that repair is based on sequence homology and that the repair products resemble those observed during an infection. Their results also suggest a RAD51-independent recombination mechanism. Many of these conclusions had been suggested previously, however this is the first time these models have been systematically tested through the recovery of large-scale observations using a high throughput system. The ability to trigger this process "on demand" provides a very powerful tool for future studies.

Specific question for the authors to consider:

1. The initial set of experiments employed the parasite line EATRO1125, which express the VSG AnTat1.1. For readers unfamiliar with this line, it would be good to know if these parasites are able to switch to expression of an alternative expression site or if they have been "fixed" to

expression only this locus (as is occasionally done through the insertion of a selectable marker within one of the ESAGs). This is relevant given that previous studies have described expression site switching in response to double strand breaks.

We thank the reviewer for this suggestion, and we agree that it is important to draw attention to this point for the reader. We have clarified this for the initial experiments in lines 157-165:

“To investigate this, we engineered tetracycline-inducible Cas9 expressing EATRO1125 parasites, which express the VSG AnTat1.1 from the natural active expression site, and induced breaks across the AnTat1.1 coding sequence using a set of guide RNAs.”

For experiments in the EATRO1125 line, there is no selectable marker upstream of the VSG to maintain expression of the active BES. For our experiments in the Lister427 line expressing AnTat1.1, there is a selectable marker (HYG) downstream of the VSG. However, we remove the drug at the start of the experiment, upon Cas9 induction, so these cells should be capable of transcriptional switching. This was not specified in the main text, but it was detailed in the materials and methods (lines 897-900). We have now updated the main text to include this detail as well in lines 285-288:

“Here, a hygromycin resistance marker is downstream of AnTat1.1. We then induced breaks in AnTat1.1 as before, removing hygromycin selection as Cas9 was induced, and detected hundreds of recombination events in two independent clones (L1-A1 = 398, L1-A2 = 833; 3C)”

2. I am curious how efficient the system is in creating DNA breaks. On lines 207-214 the authors describe obtaining 29 mosaic-expressing clones. How many clones were isolated and what percentage were not expressing a mosaic vsg? In those not expressing a mosaic vsg (if there were any), what were they expressing?

We thank the reviewer for drawing attention to this point. It is difficult to confidently estimate efficiency of break induction in this system, but our data suggest that transient electroporation of a guide is relatively inefficient, with a small percentage of parasites undergoing recombination. When a guide is incorporated into the genome and cutting is induced continuously, however, all cells appear to die or undergo recombination, suggesting relatively efficient break induction. In the revised manuscript, we provide new data to address the question of whether switching or mosaic formation occurs more often post-break in supplemental Figure 3C. We find that, under these experimental conditions, all surviving parasites express a mosaic VSG after break induction; we observe absolutely no switching. Text addressing this point has been added in lines 222-223:

“We never observed VSG switching following guide induction, only mosaic formation (Supplemental Figure 3C).”

3. Similar to point 2 above, on lines 236-245, the authors describe the results of inducing breaks in VSG-2. No mosaic vsgs were observed and the clones were no longer expressing VSG-2. Had

these parasites undergone expression site switching or had there been a recombination event that replaced the entire VSG-2 coding region?

We did not evaluate this directly, because our primary interest in this study was on mosaic recombination mechanisms. However, a contemporaneous study from the Siegel lab evaluated this and found a mix of transcriptional and recombination-based switching after a break in VSG-2, though recombination-based switches predominated by day 6 post-induction (Keneskhanova, et al., PMID: 40074895).

4. The authors frequently express surprise that the parasites were able repair a double strand break using homologous sequences from elsewhere in the genome, implying a “sophisticated homology search mechanism” and that this was “particularly unexpected”. I gather that the authors expected that the typical organization of the vsg repertoire would play a role in homology search. However, this type of homology search is quite typical for homologous recombination in virtually all eukaryotes, thus many readers will not find this in any way unexpected. If this is a truly surprising find, some additional elaboration would be helpful.

We appreciate the reviewer’s perspective on this issue. We did, indeed, expect the structure of the VSG archive to play a role in donor selection, and we were surprised to find that this was not the case. The point that sensitive homology searches are not unusual in nature is a fair one, however. In the revised manuscript, we have shifted our language to focus on the finding that the structure of the VSG archive, with its conserved repeats and subtelomeric positioning, does not appear to play a major role in mosaic recombination. We mention this in the discussion on lines 498-500:

“Therefore, the 70bp repeats are probably not driving the initial homology search following a break within a VSG and the structure of the VSG archive plays a much less prominent role in mosaic formation than we anticipated.”

5. The authors report that they observed more mosaic VSGs in parasites obtained from extravascular spaces and from this they infer that mosaics might form more frequently in this niche. Is this stated correctly? It seems equally or more likely that they form at the same rate in the extravascular spaces but that they are cleared less efficiently.

We apologize for the confusion! We believe either could be the case, and that both delayed clearance and increased recombination could play a role. We have edited this section to clarify the most salient point, which is that there are more mosaics present in extravascular spaces. These changes were made on lines 407-408 and 413-414:

“Given that parasite clearance is delayed in extravascular spaces, we wondered if mosaic VSGs may be more prevalent in this parasite niche.”

“These data suggest that mosaic VSGs may accumulate within extravascular spaces, possibly due to the slower VSG-specific parasite clearance in these spaces.”

6. In the Discussion section, the authors suggest that possibility that they are observing RAD51-independent recombination during the creation of mosaic VSGs. If this process is in fact not typical homologous recombination, is it possible to check to make sure that the donor sequence remains fully intact, as would be expected for a typical gene conversion event?

Thank you for raising this important point. New experiments have shown, to our surprise, that mosaic formation is RAD51-dependent; these new experiments are in Figure 5 and discussed in lines 341-356. While the comment's relevance to RAD51 independence may be moot now, we agree that this suggested experiment to evaluate donor maintenance would provide important mechanistic insight into mosaic formation. We therefore also addressed it experimentally in the revised manuscript using our *in vitro* system. Because of a need for sufficient drug resistance markers, we shifted our system into the new Lister427 single marker line containing T7, the Tet repressor, and Cas9 under one marker (generated by Jack Sunter and available from BEI, diagrammed in Supplemental Figure 8A). We generated transgenic parasites in this background containing AnTat1.1 in the active expression site (replacing the active VSG), a silent copy of VSG-228 donor sequence in the rDNA spacer, and a guide targeting position 694 in the rDNA spacer. Cas9 in this line is constitutively expressed, so cutting starts immediately upon guide insertion. Parasite clones were isolated as soon as possible after guide insertion (either after initial plating or a second plating if the initial population was polyclonal). After isolating genomic DNA from VSG-228 mosaic-expressing parasite clones, we amplified and sequenced the inserted VSG-228 donor cassette. The donor VSG was unaltered in every clone, suggesting this is a templated gene conversion event, as we hypothesized. This experiment is described in the text in lines 399-405:

“Finally, we used this system to further confirm that mosaic formation is a templated gene conversion event and not the result of crossing-over. Using J1339 Cas9 TetR T7RNAP Lister 427 parasites expressing AnTat1.1 containing an inserted VSG-228 donor and 694 guide in the genome, we isolated mosaic-expressing parasite clones after break induction, selected clones which utilized VSG-228 as the donor and evaluated the silent VSG-228 donor by nanopore amplicon sequencing (Supplemental Figure 8A). In all cases, the donor VSG was intact in the genome after mosaic formation, demonstrating that during mosaic formation the donor VSG serves only as a template for DNA repair.”

Referee #2 (Remarks to the Author):

Trypanosoma brucei can maintain an infection of large mammalian hosts for years despite being constantly exposed to the host immune system. The persistence of an infection is facilitated by the evolution of a population survival strategy based on antigenic variation of the VSG protein. The adaptive immune system is able to kill trypanosomes once there is sufficient VSG antibody titre but a small number of trypanosomes have switched VSG and, if antigenically novel, escape killing until recognised by antibodies in turn. At the level of the genome, VSG switching can occur through an epigenetic mechanism in which one VSG promoter is silenced and another activated, but this appears restricted to the first few weeks of an infection. The majority of switches occur through gene conversion using a reservoir of VSG sequences that comprise >25% of the genome. The gene conversion events are not simple cassette switches analogous to the budding yeast

mating type switch but are complicated usually involving several donor sequences and can generate novel VSGs. This is important as there may not be sufficient VSG coding sequence in the genome to sustain multi-year infections. The low frequency of these gene conversion events mean that the molecular processes involved have been extrapolated from before and after pictures.

In this manuscript, the authors have developed a method to systematically capture individual gene conversion events after introducing a double stranded break in the active VSG gene. The subsequent gene conversion events were found to be similar to those occurring over a time course in long term infections in mice. The experiments have allowed the authors to determine the length of the gene conversion, the length of homology required, and a positional bias.

The sequencing strategy used is novel and effective. The originality and significance of the work is that it provides a holistic view of the gene conversion events occurring with and without immune selection to single base resolution, this is the first time such events have been described in such detail.

The conclusions are generally robust. There is one point that should be discussed and that is whether antibodies just recognise the top of the VSG. It may be worth including thoughts on PMID: 8821936 in the discussion.

We have added mention of this paper to our discussion. Mounting evidence, particularly from the lab of Nina Papavasiliou, suggests that the top of the VSG is primary target of anti-VSG immunity (PMIDs: 36943866, 29988048). Our recent analysis of VSGs expressed in patients also showed the most significant divergence in VSG sequences at the top of the protein, suggesting increased selective pressure, presumably from antibody, in this region (PMID: 36354333). Similarly, our recent preprint on the targets of anti-VSG antibodies in patients also highlighted the important (but not exclusive) role the top of the VSG plays in antibody recognition (PMID: 41040341). We believe that this is likely why such small changes in the top of the VSG confer substantial immune evasion. We have now stated this more clearly in the discussion on lines 542-544 and 546-551:

“Indeed, even very small changes within the top of the VSG N-terminal lobe can confer substantial immune evasion, and mounting evidence^{49,50} suggests that the host antibody response primarily targets this part of the VSG.”

“While some previous work in vitro has shown that small changes to a VSG can affect the binding of monoclonal antibodies^{12,20,51}, most analyses of mosaic VSGs have shown antigenic similarity, as measured by cross-reactivity with polyclonal antisera, between divergent VSGs (68%-80% aa identity)^{7,51}. The fact that mosaic VSGs sharing $\geq 97\%$ similarity, as we observe here, can be antigenically distinct, suggests that the location of amino acid changes is critical to evasion.”

Overall this is a complete set of experiments that provide important new information on the mechanisms of gene conversion in VSG antigenic variation. It is exceptionally well written and is a model of clarity.

We thank the reviewer for these kind comments.

Some minor points

Line 103

'two highly repetitive and damage prone stretches of DNA'

The references (2 and 18) probably show they are repetitive but do they show they are damage prone? Is it any more damage prone than an rRNA locus?

We apologize for the confusion, the references addressing the damage-prone nature of the 70bp repeat and the telomere were at the end of the next sentence. We have moved the references in the revised manuscript to better reflect the evidence for this statement:

"It is plausible that DNA damage and repair may play a role in either mechanism, as expressed VSG genes sit between two highly repetitive^{2,21} and damage-prone^{22,23} stretches of DNA, the conserved 70bp repeat and the telomere."

Recent BLISS analysis of endogenous DNA breaks reveals the rRNA spacers as a location particularly prone to damage (PMID: 37988471), like in other species in nature. Because there are more rRNA sequences than expression sites, determining whether the rRNA loci or expression sites are more prone to DNA damage on a per bp level may be tricky; nonetheless, the most prominent BLISS signal in the nucleus is found in the single active expression site, suggesting it is a particular target for damage (PMID: 37988471). Please note, however, that BLISS mapping here suggests a complex break, with the strongest signal in the VSG ORF, not the 70bp repeats. Either way, our primary goal in this paragraph was to highlight the plausibility of DNA damage playing a role in mosaic formation. Therefore, the amount of DNA damage occurring in the rRNA locus vs the expression sites should not influence the major findings of the study.

Line 209

A brief explanation of why 7 days induction was used?

It takes approximately 7 days to isolate colonies from single cells, and our goal was to isolate mosaic-expressing clones from single cells. As we were uncertain of the efficiency of break induction, we chose to maintain induction for the entire time to ensure efficient mosaic induction. We have edited the text to reflect this explanation on line 218-220:

"Using parasite lines that stably express a VSG-targeted guide RNA, we induced Cas9 expression for 7 days to allow mosaic-expressing parasites to form from individual cells (2C; Supplemental Figure 2A)."

Line 496

A comment of whether the NHEJ pathway absent in the majority of trypanosome species? Especially those that do not use VSG based antigenic variation?

NHEJ is absent in the majority of trypanosome species, even those who do not undergo antigenic variation. However, this paragraph has been edited substantially in response to our new data reflecting the RAD51-dependence of mosaic formation. We no longer mention NHEJ as we now know that mosaic formation is more likely occurring through a form of RAD51-dependent HR.

Referee #3 (Remarks to the Author):

Overview:

Several pathogens, including African trypanosomes undergo antigenic variation to evade the host immune response and rely on genomic diversification to extend their antigenic repertoire. Many studies have demonstrated that novel VSGs are generated during infections, particularly in late stages of infection. These are thought to emerge through segmental gene conversion ('mosaic' VSGs) or de novo point mutations.

Indeed, the contribution of VSG pseudogenes to the formation of novel 'mosaic' VSGs has long been known (PMID: 15463711; PMID: 1380125; PMID: 2574459). In my view, the scale and complexity of the segmental gene-conversion events associated with the generation of functional 'mosaic' VSGs are clear (PMID: 17652423; PMID 23853603; PMID: 25814582; etc). However, what remains elusive is how this is catalysed.

Studying such mechanisms has been incredibly challenging due to the lack of an in vitro system to do so. Therefore, I congratulate the authors (Smith et al.) on the generation of an efficient and reproducible in vitro system to study this process in a controlled manner, which will no doubt enable hypothesis-driven exploration of mechanisms underpinning antigen diversification in *T. brucei* (and potentially other pathogens).

However, I am not convinced that as it stands this work represents a groundbreaking discovery, since the DNA repair mechanisms involved in mosaic antigen formation remain elusive. I find that most of the observations are consistent with previous work, which at times seems insufficiently credited, and do not add significant mechanistic insight.

Please find specific comments below.

Major Points:

1- The authors state that the frequency of mosaic formation decreased when moving further away from the centre of the AnTat1.1 coding sequence and excluded that this could be due to the guide cutting efficiency based on the results depicted in Fig. S2H. There is a high variability for some of the gRNAs, therefore the data is not very compelling and raises the question whether some of

the differences can indeed be due to different guide cutting efficiency. Also, can anti gamma-H2A expression be used as a reliable measurement for guide cutting efficiency? What is the evidence for this?

Gamma-H2A has been used extensively as a marker for DNA damage in *T. brucei* (see PMIDs: 31289179, 40074895, 23555264). Gamma-H2A signal varies across the cell cycle, and its higher levels during S and G₂ are correlated with the increased levels of DNA damage occurring during DNA replication (PMID: 31289179). Signal also increases following damage by ionizing radiation, with a loss of the protein as damage is repaired (PMID: 29599445). While it is likely an imperfect assay, evidence suggests this should serve as a reasonable proxy for DNA damage in a cell population.

Transient electroporation of the guide did not produce enough sustained DNA damage for detection of gamma-H2A signal by immunoblot, and so we instead inserted the guide into the genome to attempt to assess guide efficiency. Here, we generated multiple independent clones due to the known position effects within the possible rDNA spacer sites which could impact expression levels of the guide (PMID: 16182389). We induced guide expression for 24 hours and our immunoblot analysis of gamma-H2A showed induction in each clone except for guide 243. While we agree with the reviewer that the variability is not ideal, guide 694, which produces orders of magnitude more mosaic VSGs than the other guides under transient electroporation, does not induce an extreme gamma-H2A signal. In fact, gamma-H2A signal for guide 694 is rather modest compared to guides 369R and 978R.

In order to reduce potential positional effects from variable insertion, we plated parasites under limiting dilution such that a single parasite was induced to form recombinant VSGs with a control, uninduced plate to determine the number of parasites that died, which we believe can be used in combination with recombination as a proxy for cutting (Supplemental Figure 3B). Because we are inducing sustained cutting, potential differences in guide expression level should play less of a role in this assay. We observe no significant difference between the rate of parasite death post induction between all the clones. We also made another observation that may help explain, in part, our increased recombination at position 694. Many of the 243 and 1459 guide-expressing parasites form mosaic colonies with very small minority populations of mosaic VSGs (predominant species is the original VSG). This suggests that these recombinants might not grow as well as those from 694 supporting our hypothesis that the VSG is better able to tolerate differences within the unstructured apex of the VSG.

While we believe the evidence for this interpretation is strong enough to warrant discussion in the manuscript, we agree that it is not without gaps. Therefore, we have made sure that the manuscript highlights this region of the VSG as a *possible* recombination hotspot, not an undeniable feature of mosaic VSG recombination. We also tuned the language to refrain from claiming “formation” is increased here over other spots but focus on successful generation of events and overall mosaic prevalence following a cut. These changes are found on lines 208-210, 211-214, and 424-425:

“Notably, as the DNA breaks progressed further away from the center of the AnTat1.1 coding sequence, the frequency of observed mosaics decreased dramatically (2B).”

“Rates of parasite death after induction of DNA breaks also did not vary significantly between guides, further suggesting guide cutting efficiency did not account for differences in the number of mosaics detected at different break positions (Supplemental Figure 3B).”

“We specifically utilized mosaic recombination derived from guide 694 induced breaks as those were the most prevalent in our assays.”

2- Following on point 1, all the experiments were performed with AnTat1.1, for which mosaics were reported as far back as 1983 (Pays et al, Cell, 1983, PMID 6616615). Have you tried to look at other VSGs that are members of VSG families? Do you obtain the same pattern of recombination events *in vitro* and *in vivo*?

We agree with the reviewer that it is important to show this phenomenon holds true for VSGs in general and is not AnTat1.1-specific. To that end, we isolated two additional VSGs-expressing parasites in our Cas9-expressing EATRO1125 line, Tb427VSG-8 and EATRO1125VSG-73, which we used to explore mosaic formation (note: Tb427VSG-8 is found within the EATRO1125 genome, but was absent from EATRO1125 VSGnome, which is why it is referred to as Tb427VSG-8 in the manuscript and not with an EATRO1125 VSG ID). After inserting guides into the genome and inducing double stranded DNA breaks *in vitro*, we observe the same pattern of short insertions, templated by closely related family members, that we previously observed in AnTat1.1. We also used VSG-8 expressing parasites to infect μ MT mice to confirm that we could observe short insertions *in vivo* in a second VSG. Using nanopore sequencing, we observed numerous short insertions which increase in prevalence over time during the infection, mirroring the results we observe with AnTat1.1. These new experiments demonstrate that the patterns we observed in AnTat1.1 are a common feature of mosaic formation and not specific to that long-studied VSG.

These new results are found in Supplemental Figure 4 (line 1700). They are discussed in the main text at lines 244-249 and 388-390:

“We observed the same short insertions when breaks were induced in the coding sequence of two other VSGs with homologous family members, Tb427VSG-8 and EATRO1125VSG-73, suggesting the observed patterns are not specific to AnTat1.1 mosaics (Supplemental Figure 4A-4D; while Tb427VSG-8 is in the EATRO genome, it is only annotated in the Lister427 VSGnome² and therefore lacks an EATRO VSG ID).”

“We also observed similar short mosaic insertions in Tb427VSG-8 during μ MT infections (average = 70.5 bp, median = 42 bp) using nanopore sequencing (Supplemental Figure 4E-4G).”

3- As mentioned above in the overview paragraph, the scale and complexity of the segmental gene-conversion events that underpin the generation of functional ‘mosaic’ VSGs are clear (PMID: 17652423; PMID 23853603; PMID: 25814582; etc). However, what remains elusive is the molecular mechanism. The authors have developed an efficient and reproducible *in vitro* system to study this process in a controlled manner, but their observations do not extend much beyond what was already known apart from the very interesting observation that donor VSGs can be used as templates irrespectively of their genomic localisation. In my opinion, a real breakthrough would

have been determining the DNA repair mechanisms underpinning mosaic formation, which the authors do not demonstrate.

We believe the revised manuscript provides substantial mechanistic insight into mosaic formation. Exactly as the reviewer mentioned, our data are consistent with many previous observations and even previous predictions, but critically, none of these were testable before the development of the tools presented in this manuscript. We define the following mechanistic features of mosaic formation in the paper:

- **Mosaic formation is templated.** It has been long hypothesized that mosaics form via templated gene conversion events, but there was no experimental system to test this hypothesis. Using our new system, we show experimentally that mosaic formation is the result of a templated gene conversion event—in the absence of a template, mosaics will not form. In the revised version of the manuscript, we have added data showing that an ectopic template used for mosaic recombination remains unaltered after recombination, further emphasizing that this is a templated event and not a crossing-over.
- **Genomic position does not influence mosaic formation.** We appreciate that the reviewer found this result to be interesting. It had been hypothesized that the genomic position of a donor VSG would matter for VSG recombination, but our results unequivocally demonstrate that characteristic features of the VSG archive like the VSG UTR, subtelomeric position, and the 70bp repeats are not essential for mosaic formation.
- **Mosaic formation requires very little homology.** Our revised manuscript includes an analysis of the minimal homology required for mosaic recombination. We show that only 50bp of flanking, *imperfect* homology is required for a mosaic donor template to be used. 100bp of flanking homology allows recombination at an efficiency similar to the full-length VSG. This is the first experimental evidence for the sequence requirements of mosaic formation.
- **Mosaic formation is RAD51- and BRCA2-dependent.** We thank the reviewer for pushing us to investigate the machinery involved in mosaic formation. In response to this comment and the one below, we performed our experiments in RAD51- and BRCA2-knockout parasite lines. To our surprise, we find that mosaic recombination is dependent on both proteins, suggesting a homologous recombination (HR)-like pathway is driving these recombination events. This, we believe, substantially increases the impact of the paper, as it represents the first identification of cellular machinery involved in mosaic formation. Moreover, it provides data in stark contrast to the prevailing model of mosaic formation, which hypothesized that an MMEJ-like mechanism drives the process. We are grateful to the reviewer for allowing us to set the record straight on these mechanisms before publication of the manuscript.

In summary, we believe that in addition to presenting a critical new toolkit for studying an important immune evasion mechanism, this paper provides an enormous amount of mechanistic information regarding mosaic VSG formation. The surprising observation that very short imperfect homologies underlie this RAD51-dependent HR-like mechanism further increase the work's impact, as it highlights features of HR in *T. brucei* that were not previously appreciated.

Smith et al speculate that the recombination events they describe are likely RAD51-independent, but have they tested this in RAD51 KO cells? An interesting possibility is microhomology-mediated end-joining (MMEJ), which has been previously suggested to contribute (Glover et al, NAR, 2011, PMID: 20965968), and the authors also mention this in the discussion. Why not test this?

The revised manuscript now includes this experiment, as well as an evaluation of the dependence of BRCA2 on mosaic recombination. To our surprise, the formation of mosaic VSGs is RAD51-dependent. These data are presented in Figure 5 and Supplemental Figure 8B, as well as on lines 341-355:

“Given the homology requirements of mosaic recombination, we wondered whether factors known to be involved in homologous recombination are required for break-induced mosaic formation. Both RAD51³² and BRCA2³³ are known to play a role in DNA repair and antigenic variation in T. brucei. In addition, microhomology mediated end-joining (MMEJ), which relies on ~5-20 bp of imperfect homology³⁴, comparable to that which we observe flanking mosaic recombination sites, has been described in T. brucei and occurs independently of RAD51³⁴⁻³⁶. To test the role of RAD51 and BRCA2 in mosaic recombination, we transiently transfected amplicons encoding sgRNAs targeting position 694 in Cas9-expressing parasites lacking either RAD51 or BRCA2 (Supplemental Figure 8B). Mosaic recombination was dramatically reduced in the absence of BRCA2 and completely abolished in the absence of RAD51 (5A & 5B). Together, these results suggest that break-induced mosaic recombination requires the homologous recombination machinery RAD51 and BRCA2 to operate.”

4- The authors make the bold statement that ‘small changes in the VSG sequence provide substantial antibody evasion’ (line 351) and ‘substantial immune evasion’ (lines 457-458). However, what they do show is reduced antibody binding, not immune evasion, which by their own admission in the discussion might not be sufficient. I quote “However, a 50% drop in binding may not be sufficient to fully evade preexisting anti-VSG antibody.” Can you show that these differences in antibody binding lead to immune evasion?

We would argue that immune evasion exists on a spectrum, and proving immune evasion would therefore be extremely difficult. For example, a mutation that is evasive at a timepoint when anti-VSG antibody titers are lower may not be evasive at a later timepoint when titers are higher. The survival conferred by one recombinant at an earlier, lower-titer point, may provide enough time for that parasite to undergo a second, similarly evasive mutation/recombination. This is why we believe our observation that more mosaics can be detected in tissues is particularly important: overall immune pressure in extravascular spaces appears to be lower than the bloodstream, potentially allowing for the iterative recombination that generates the complex mosaics observed in circulation later in infection. Therefore, we would argue that even incomplete evasion is likely to play a role in infection dynamics and parasite survival.

We do agree, however, that our assay, using a very non-physiological aliquot of purified anti-AnTat1.1 rabbit serum, was less than ideal. To further interrogate the physiological relevance of the mutations we observe, we analyzed the ability of mouse serum collected during infections with AnTat1.1-expressing parasites to bind mosaic AnTat1.1 mosaic derivatives. These

experiments reveal how variable the anti-VSG response can be in different infections, further highlighting how tricky true immune evasion can be to define. Nevertheless, these experiments, now depicted in Figure 7C, show even larger drops in binding, up to 75%, for some mosaics. We believe this addition significantly strengthens our argument that these variants would provide some immune evasion in the context of infection.

We address these this new experiment on lines 426-431:

“We also evaluated binding of mouse antisera collected after infection with AnTat1.1-expressing parasites, to better understand how mosaic VSGs might evade antibodies in a more natural context. Here we observed even larger decreases in binding for certain mosaics, although some variants that showed evasion of post-immunization rabbit AnTat1.1 antisera did not evade serum from mouse infections (7C).”

Also, how are these observations novel or different from what has been previously reported by others, among those Hall et al, PLoS Pathogens, 2013, PMID 23853603 (see Figure 6 specifically)?

There are two key differences between this observation in our manuscript and the Hall paper. The first is that the variants we analyzed were obtained agnostically, in the absence of antibody pressure. In Hall, et al, mosaics were chosen from an 8-day period during infection, a timepoint at which each variant had, very likely, survived exposure to large amounts of pre-existing host antibody. In our study, we analyzed the evasive capacity of the most common outcomes after recombination. While the experiments are undeniably similar, we believe that our analysis provides additional nuance by revealing how evasive individual recombinants can be when isolated in the absence of antibody selection.

The second key difference is in the nature of the mosaic VSGs investigated in each study. The variants we are measuring are highly similar to the original VSG AnTat1.1 (~97%-99.4%) and each was generated following a single insertion event by a single donor VSG. The mosaic VSGs Hall et al. isolated for serological assessment were very dissimilar to one another (78%-87.5% within the N-terminal Domain) and arose from 4 donor VSGs and numerous segmental gene conversion events. It is also worth noting that the VSG repertoire of TREU927 strain used in this study was not well-characterized at the time, so recombination sites had to be inferred from an incomplete repertoire and largely remained tentative. All of the Hall mosaics analyzed were serologically similar except for a single antigenically distinct variant. While Hall et al. points out that 22 of the amino acid changes found in the evasive variant are within the “membrane-distal end” of N-terminal loops of the VSG, these changes appear to occur throughout the top half of the VSG N-terminal domain. Because we evaluate smaller changes with fewer donors, our study is better able to highlight more precisely the regions of the VSG protein most likely to be critical for evasion. Notably, Hall, et al, even point out that this is an area for future research, suggesting it remains “to [be] establish[ed] how combinatorial variations generated by SGC [Segmental Gene Conversion] can yield antigenically unique variants.” While we remain appropriately tentative in our conclusions, we believe our results refine our understanding of how the antigenically distinct variant in the Hall paper achieved immune evasion: our work suggests that the mutation of the residues at the top of the VSG—specifically where we observe increased recombination and

immune evasion—allowed this variant to become immune evasive while the other VSGs, which primarily varied at other sites, remained antigenically similar.

Citation and/or discussion of other publications:

1- In the introduction you mention “While mosaic VSGs have been observed in the literature, they are often found under extreme experimental conditions...” – can you define ‘extreme experimental conditions’?

We agree with the reviewer that this could have been more precise. We have edited the text to reflect our intended meaning on line 94-96:

“Where mosaic VSGs have been described in the literature, they are often found under strong, antibody-mediated selection^{11–15} or late during infection^{5,7,16,17} making it difficult to discern how exactly they arose.”

2- Also in the introduction, the authors mention that ‘DNA breaks are frequently observed within the 70bp repeats at the active and near telomeres within silent BESs’, but since they induce breaks in the VSG ORF, this paper is relevant: Girasol et al (PNAS, 2023, PMID: 37988471) show naturally occurring breaks at the VSG-2 ORF in BES1 L427 (repaired by RAD51) using BLISS.

Thank you for pointing this out. The introduction has been edited to reflect this suggestion on lines 107-110:

“Locus-directed mapping has detected DNA breaks within the 70bp repeats at the active expression site²² and near telomeres within both the active BES²³ and silent BESs²³, while genome-wide DNA break mapping has shown highly abundant, complex breaks within the expressed VSG and confined to the active BES²⁴.”

3- When the authors conduct experiments in L427 (Fig. S3), how does the observed switching pattern align with the original experiments conducted by Escrivani et al (PMID 37459347)? This is a paper the authors do not cite and that uses the same approach to trigger VSG switching in L427.

This was an unintentional omission, and we thank the reviewer for pointing this out! We have edited the introduction to include a reference to this paper on line 111-113:

“Experimental evidence implicates DNA damage in VSG switching more generally, as a DNA double-strand break induced upstream of the VSG^{22,23} and within the first 15 bases of the VSG-2 CDS²⁵ induces a switch.”

Like in Escrivani et al., we also observe that all activated VSGs are expression-site associated. We have included this in the results on lines 257-260:

“Parasite clones isolated after a break in VSG-2 no longer expressed VSG-2, and the VSGs expressed by these clones, which were all expression site-associated, showed no evidence

of mosaic recombination (Supplemental Figure 5B; cut position 707, n = 4; cut position 1082, n = 3).”

4- I think there are several papers that although were cited in the introduction, should have discussed, a striking one being: Hall et al, PLoS Pathogens, 2013, PMID 23853603. Although the in vitro method and the combination of both in vitro and in vivo experiments are overall commendable, it is not entirely clear what are the novel findings in this paper in relation to previous studies.

We believe this manuscript represents a significant advance from the previous work by Hall et al. In addition to the enormous mechanistic insight the revised manuscript has provided (detailed in response to Point 3 by this reviewer), we would like to make a few additional points about how this manuscript is additionally novel with respect to Hall et al.

- **Evidence for “*progressive mosaicism*”** Hall et al. provides two competing hypotheses for mosaic formation: progressive mosaicism and rapid mosaic generation. They suggest, given their limit of detection, that substantial evidence for iterative mosaic VSG formation is lacking. Here we find that, the vast majority of the time, a blunt, double-stranded DNA break leads to a single insertion event from one donor VSG. We observe this phenomenon with 3 distinct VSGs suggesting that accumulation of insertion events following DNA damage is what generates the complex mosaic VSGs observed late in infection. Further, in μ MT mice, we observe an increase in mosaic VSGs over time, suggesting that insertions accumulate throughout infection. While this does not rule out a rapid mosaic generation mechanism, it provides substantial support for an iterative hypothesis. This is detailed in the discussion on lines 555 to 565.
- **Evidence for the “*efficiency*” of immune evasion** Hall, et al., also express curiosity about the inefficiency of the production of many serologically similar mosaic VSGs from numerous donor VSGs. Here, we connect our observation that mosaic VSGs form more readily within the unstructured regions of the apex of the VSG N-terminal domain with their immune evasiveness, suggesting that parasites may have mechanisms to bias small changes towards immune evasion.
- **Evidence that “*incomplete variation*” may contribute to complex, natural infections** Within the discussion, Hall, et al., suggest that isolation of many serologically similar mosaic VSGs may be a consequence of decreased immune capacity at later stages of infection. As stated in response to point 4 for reviewer 3, here we demonstrate that in the tissue spaces, where immune pressure is reduced compared to the blood, we observe more mosaic VSGs. This suggests that natural infections allow partially evasive variants to persist within tissue spaces, perhaps until full immune evasion can be achieved.

While the Hall paper was an important initial characterization of VSG expression during infection and clearly highlights the critical role of mosaic VSGs during chronic infection, we believe our extensive mechanistic characterization of mosaic VSG formation and ability to address some of the outstanding questions raised by Hall, et al., represent a substantial advance, building significantly on the findings of the Hall paper.

Other Points:

1- In the M&M, the authors state that “Mosaic reads which did not overlap to generate a consensus and did not contain an identifiable recombination site in read1 were excluded from analysis.” What is the % of reads that were discarded in this process?

Thanks for this comment. The revised manuscript now includes a table with these data found in Supplemental Excel File 3. For *in vitro* experiments, only ~1.6% of reads were discarded. For *in vivo* samples, this number was higher, at ~30% on average. We are reluctant to overinterpret this data, because it can be influenced by the efficiency of the fragmentation reaction, which is dependent on RNA input. Because we could not control input material as carefully for *in vivo* experiment where RNA was extracted from whole blood, we believe fragmentation may have been less efficient, resulting in longer fragments and a lower proportion of overlapping reads. We do not believe this affects the interpretation of the data significantly; if anything, these numbers may suggest we are underestimating recombination *in vivo*. We describe this analysis in the materials and methods on lines 1154-1163:

“A summary of these is found in supplemental excel file 3. The average proportion of mosaic VSGs without an identifiable recombination site was ~1.6% for in vitro samples with at least 10 mosaic reads identified (max = ~9%). Samples derived from in vivo mouse experiments tended to have more unidentifiable recombination sites, with a majority occurring in the 0R and 1R primer samples (avg = ~30%, max = ~43%).”

2- In Fig. 2A, is the data for a single clone per gRNA or a combination of 2 clones? If it is a single clone, how reproducible are the donor VSGs between clones? Could you show the 2 clones per gRNA separately (in Supp for instance)?

The data presented were for a single clone. The visualization of data for the second clone is now included in the revised manuscript as Supplemental Figure 3A.

3- Are you sure the graph in Fig. 2G is not duplicated in Fig. S3A? Or is it just very consistent?

We appreciate this reviewer’s attention to detail and careful review of our work. We also double checked this, but the results are remarkably similar.